# Comprehensive profiling of neutralizing polyclonal sera targeting coxsackievirus B3

Beatriz Álvarez-Rodríguez [1] ✉, Javier Buceta[1] ✉ & Ron Geller [1] ✉

Despite their fundamental role in resolving viral infections, our understanding of how polyclonal neutralizing antibody responses target non-enveloped viruses remains limited. To define these responses, we obtained the full antigenic profile of multiple human and mouse polyclonal sera targeting the capsid of a prototypical picornavirus, coxsackievirus B3. Our results uncover significant variation in the breadth and strength of neutralization sites targeted by individual human polyclonal responses, which contrasted with homogenous responses observed in experimentally infected mice. We further use these comprehensive antigenic profiles to define key structural and evolutionary parameters that are predictive of escape, assess epitope dominance at the population level, and reveal a need for at least two mutations to achieve significant escape from multiple sera. Overall, our data provide a comprehensive analysis of how polyclonal sera target a non-enveloped viral capsid and help define both immune dominance and escape at the population level.

Neutralizing antibodies are key to resolving viral infections and can provide long-term protection against reinfection. These antibodies mostly target viral proteins involved in cell entry, namely membrane proteins in enveloped viruses and capsid proteins in non-enveloped viruses[1]. Consequently, vaccination strategies frequently aim to elicit polyclonal neutralizing antibody responses utilizing these same viral proteins[2]. In turn, viruses must overcome these immune responses for their successful spread in previously infected or immunized populations, establishing a continuous evolutionary arms race to alter immunodominant epitopes and refine antibody responses[3].

Recent high-throughput approaches have provided new insights into how viral membrane proteins are targeted by polyclonal antibody responses for several enveloped viruses, including human immunodeficiency virus, influenza A virus, and SARS-CoV-2[4–9]. These studies have revealed the breadth and relative strength of neutralization sites induced by both natural infection as well as vaccination and helped define mutations conferring escape from neutralization. However, our knowledge of how non-enveloped viruses are targeted by polyclonal sera remains limited, despite the fact that they constitute >40% of mammalian viruses[10]. Moreover, fundamental differences between capsid proteins and viral envelope proteins could preclude the extrapolation of results from enveloped to non-enveloped viruses. In

particular, carbohydrate modifications that alter the sensitivity of viral membrane proteins to antibody neutralization[11] are absent in viral capsids. Additionally, non-enveloped viral capsids encode multiple functions not found in viral membrane proteins, including the information for assembly, genome packaging, and genome release, which could significantly constrain their ability to tolerate mutations conferring immune escape. Obtaining a deep understanding of how viral capsid proteins are targeted by, and escape, polyclonal antibody responses is therefore of key importance for understanding host-pathogen interactions and viral evolution of this large fraction of viruses.

Picornaviruses were the first human viruses to be structurally defined at the atomic level[12], revealing an icosahedral capsid whose symmetry is conserved across all capsids of non-enveloped viruses in vertebrates. The picornavirus capsid is comprised of 60 copies of four structural proteins, three of which are surface-exposed (VP1, VP2, and VP3) and one that lines the internal capsid surface (VP4)[13]. A depression in the capsid, termed the canyon, frequently harbors residues involved in receptor binding[14]. Antibody neutralization in picornaviruses has been mapped to four surface-exposed structural regions using escape from monoclonal antibodies (mAbs) and structural studies: the canyon northern rim (five-fold axis), inner surface (canyon floor), and southern

[1]Institute for Integrative Systems Biology (I2SysBio), Universitat de Valencia-CSIC, Valencia 46980, Spain. ✉e-mail: beatriz.alvarez-rodriguez@uv.es; javier.buceta@csic.es; ron.geller@csic.es

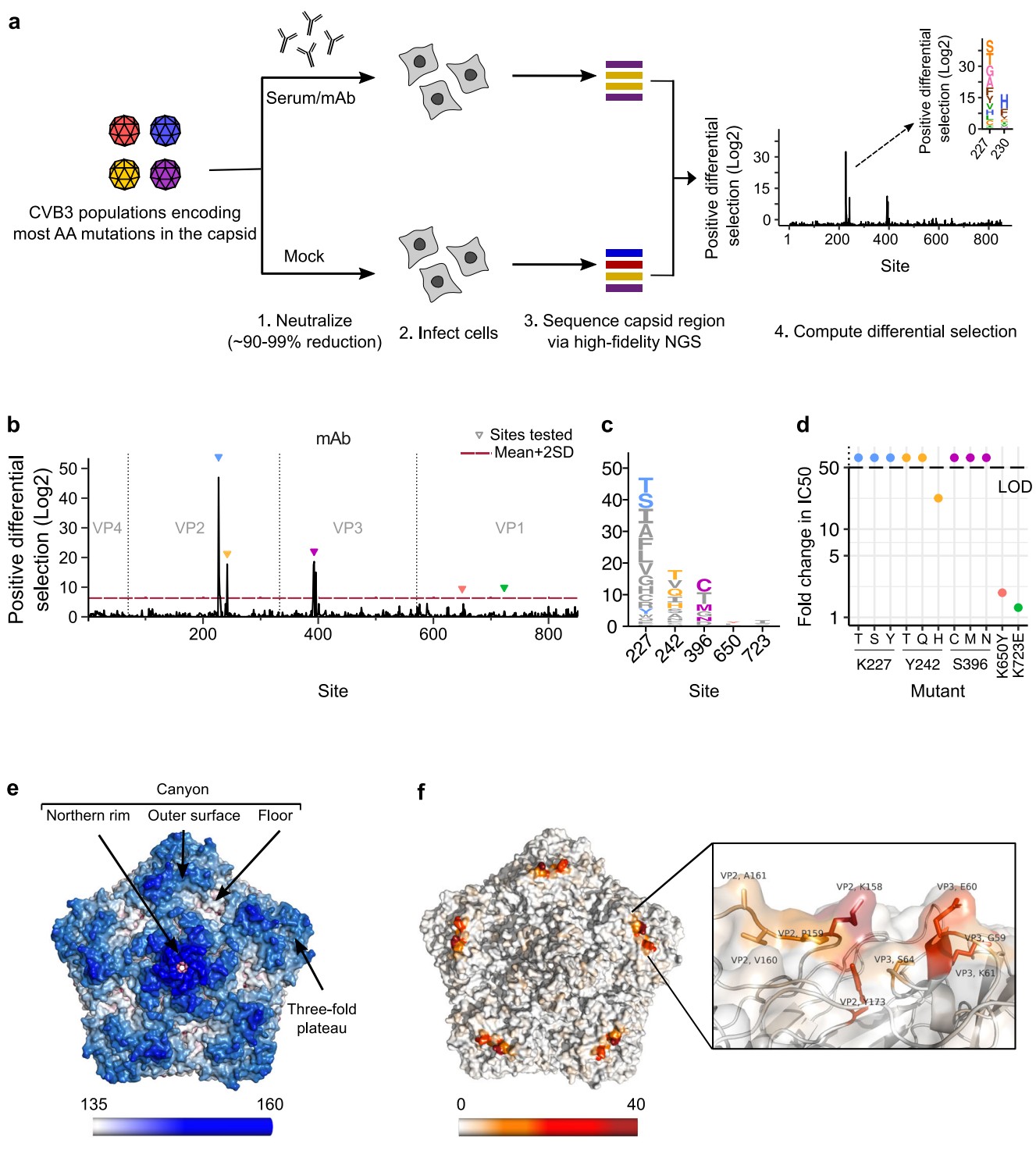

rim (canyon outer surface) as well as the two and three-fold plateau (see Fig. 1e)[15–18]. The mechanisms by which mAbs neutralize picornaviruses have also been extensively studied, and include impeding receptor binding, premature induction of genome release, and virion stabilization[17,19]. The large body of knowledge of how picornaviruses are targeted by mAbs combined with the fact that humoral responses are essential for resolving picornavirus infections[20] make picornaviruses excellent models for studying antibody-capsid interactions.

In this work, we comprehensively map the antigenic profiles of multiple human and mouse-neutralizing sera to a model human picornavirus, coxsackievirus B3 (CVB3). Our results reveal a conserved antigenic profile in mice that focuses on a single common antigenic site. In contrast, we find human polyclonal responses to be diverse, targeting anywhere from a single dominant site strongly to several sites across the capsid with reduced potency. Capitalizing on the large dataset of antibody-escape mutations defined in our study, we show that a few key structural and evolutionary parameters are sufficient to predict escape with relatively high accuracy. Finally, we define the immune dominance of the identified escape sites at the population level and demonstrate that a combination of two mutations can confer

**Fig. 1 | Mutational antigenic profiling workflow for CVB3. a** Overview of the experimental workflow. CVB3 populations harboring high diversity in the capsid region are neutralized or mock treated and surviving viruses amplified by infection of cells. Mutation frequencies across the capsid are then obtained via high-fidelity deep sequencing. Mutations showing positive differential selection, i.e. those whose frequency relative to that of the WT has increased following neutralization versus mock-neutralized controls, define mutations conferring escape from antibody neutralization. **b** Mutational antigenic profile of a neutralizing mAb. Triangles indicate sites that were experimentally validated in **d** and the dashed red line represents the mean+2 SD of all mutations showing positive differential selection. **c** Logo plot representation of sites selected for validation in **d**. The height of the letter is proportional to the positive differential selection value. **d** The IC50 fold change for the mutant versus WT virus. The dashed line indicates the upper limit of detection (LOD) of the assay, calculated as indicated in the text. **e** CVB3 capsid pentamer structure (PDB ID: 4GB3) with residues colored by distance to the center of the capsid. The four main antibody binding regions are indicated. **f** The CVB3 capsid pentamer structure colored by the positive differential selection values of the mAb antigenic profile, with a zoomed view in the right panel. Source data are provided as a Source Data file. See Figure S1 for individual replicates and the correlation between replicates, Table S1 for the surviving fraction of the viral populations after neutralization, and Table S2 for structural location of the main escape sites. See capsid_roadmaps for capsid roadmaps labeled by residue and dms_view for interactive visualization on GitHub[23].

significant evasion from neutralization by multiple human sera. Overall, our results provide a comprehensive understanding of how polyclonal neutralizing antibody responses target a prototypical human picornavirus and reveal epitope dominance at the population level.

## Results

### An experimental workflow defines mutations conferring escape from antibody neutralization across the complete CVB3 capsid

To define how picornaviruses are targeted by polyclonal antibody responses, we modified an experimental workflow that provides a quantitative evaluation of how mutations in a viral population confer escape from antibody neutralization, termed deep mutational antigenic profiling[21] (Fig. 1a). Briefly, two previously characterized CVB3 populations derived from plasmid libraries encoding 96% of all possible single amino acid mutations across the capsid region were employed[22]. Following growth in cells, non-viable and highly deleterious mutations were purged from these viral populations, resulting in 86% of all possible single amino acid mutations being represented (~13,900 mutations; 78% and 68.6% of all possible single AA mutations in the population 1 and 2, respectively[22]). To define which mutations reduce sensitivity to antibody neutralization, these populations are treated with either a monoclonal antibody (mAb) or neutralizing sera so that infectivity is reduced by 90–99% relative to mock treatment, balancing sufficient selection to observe escape without losing mutations with intermediate phenotypes (Fig. 1a). Mutants escaping neutralization are then enriched by infecting cells for a single cycle. Finally, the frequencies of capsid mutations in both the neutralized and mock-neutralized populations are obtained using a high-fidelity next-generation sequencing technique[22]. Mutations showing positive differential selection, i.e., those whose frequency increases versus the WT amino acid at a given position, indicate the contribution of the individual mutation to escaping antibody neutralization. Similarly, the sum of all positive differential selection values at a given residue reflects its overall contribution to evading antibody neutralization (termed site positive differential selection).

To validate our ability to identify mutations conferring escape from antibody neutralization, we first treated the CVB3 populations using a commercially available mouse mAb (reciprocal 50% inhibitory concentration, IC50, of 5,525; see neutralization_results on GitHub;[23] Fig. 1b and Table S1). Experiments were performed twice with each of the two viral populations. Site positive differential selection values showed good correlations between replicates of the same viral populations (Pearson's *r* of 0.72–0.81; Figure S1) but were somewhat lower between biological replicates performed with different virus libraries (Pearson's *r* of 0.67–0.69; Figure S1), as expected due to the different mutations present in each population. The main sites of escape in the mAb profile were comprised of several residues across the VP2 EF loop and VP3 AB helix/AB loop (Fig. 1f, and Table S2). These two regions on the outer surface of the canyon (Fig. 1e) have been described as antigenic sites in different picornaviruses and are known as the "puff" and the "knob"[24]. Mapping of the positive differential selection values to the structure of the CVB3 capsid locates these sites in a single region of

escape on the outer surface of the canyon, as expected for a mAb (Fig. 1f; interactive visualization available on GitHub[23]).

To validate our mutational antigenic profiling approach, we evaluated the effects of nine mutants predicted to escape neutralization (differential selection scores of 1.45–4.9) from the three most relevant sites of escape and two mutations predicted to confer low (K650Y) or no (K723E) escape (Fig. 1c; see table CVB3_master_table for polypeptide position to chain and residue conversion, available on GitHub[23]). All mutants from the three principal escape sites strongly evaded neutralization (Fig. 1d), yielding fold reductions in the IC50 ranging from 22-fold to the limit of detection of the assay (>50-fold; lowest mAb dilution tested, 1:100, divided by the IC50 for the WT virus; Fig. 1d). On the other hand, the K650Y and K723E mutations did not alter neutralization significantly (IC50 fold change <2). Together, these results validate our ability to map escape from antibody neutralization at both the site and mutation levels.

### Mouse polyclonal antibody responses are uniform and discrete

Laboratory mice are the most common model for investigating CVB3 infection in vivo[25,26]. To understand the humoral immune response in experimentally infected mice, we challenged three mice with CVB3. The sera of all mice were neutralizing at 3 weeks post-challenge, with reciprocal IC50 values of 1863, 322, and 572 for samples m1, m2, and m3, respectively (see neutralization_results on GitHub[23]). We next evaluated the antigenic profile of these sera. Unexpectedly, all three sera showed a similar antigenic profile, suggesting that the response of experimentally immunized mice is relatively invariable (Fig. 2a and Figure S1). The strongest escape mutations in these antigenic profiles were located in several residues across the VP1 protein, focused within a single structural area of escape in the northern rim of the canyon, involving the EF and BC loops of VP1 and residue Q264 in the C-terminal loop of VP1 (Fig. 2d and Table S2). Residues within these loops have been previously defined to interact with mouse-neutralizing mAbs in several enteroviruses[27–29], suggesting a conserved response in mice.

We next evaluated whether mutations in two of the main sites of escape (K650Y and K723E, Fig. 2b) affected neutralization by the mouse sera. Indeed, both mutations resulted in a nearly complete loss of neutralization at the lowest serum dilution tested (1:100), precluding the calculation of their IC50 (Fig. 2c). To obtain a relative quantification of how these mutations altered neutralization, we defined the IC50 fold-change for each serum as greater than the limit of detection of our assays as for the mAb. For the most neutralizing mouse sera (serum m1), a >10-fold reduction in sensitivity to neutralization was observed, although the actual value could be significantly higher (Fig. 2c). Overall, these results reveal the immune response of experimentally infected mice to be uniform and targeted to a delimited region of the capsid.

### Antigenic profiles of human sera are highly variable

Enteroviruses, such as CVB3, cause frequent infections in humans[30]. The humoral response plays a key role in resolving these infections

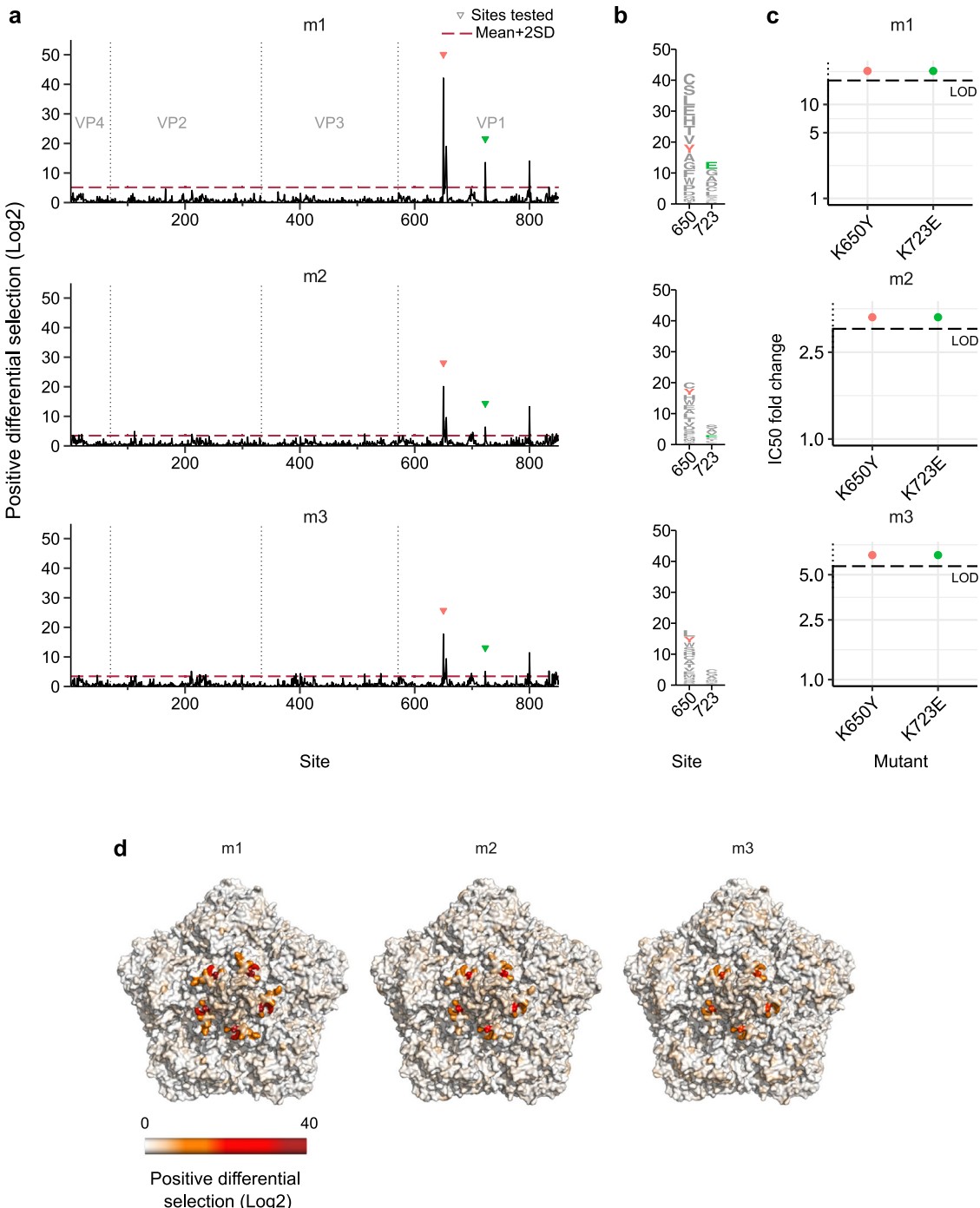

**Fig. 2 | Antigenic profiles of sera from CVB3 infected mice. a** Positive differential selection profiles of sera from mice infected with CVB3. Triangles indicate sites validated in **c**. **b** Logo plot representation of sites of escape selected for validation, with mutations evaluated in **c** colored. **c** IC50 fold change for the mutant virus versus the WT virus. **d** The CVB3 capsid pentamer structure colored by positive differential selection values of the mouse antigenic profiles. See Figure S1 for individual replicates and the correlation between replicates, Table S1 for surviving fraction of the viral populations after neutralization, and Table S2 for structural location of the main escape sites. Source data are provided as a Source Data file. See capsid_roadmaps for capsid roadmaps labeled by residue and dms_view for interactive visualization on GitHub[23].

and confers protection from reinfection[19,20,31]. To define how polyclonal human sera neutralizes the CVB3 capsid following natural infection, we first evaluated the ability of 140 sera from healthy adult donors to neutralize the CVB3 Nancy lab strain (see biobank results on GitHub[23]). Overall, 60% of sera was neutralizing (defined as a reciprocal IC50 > 40; n = 85/140, Figure S2a), with higher prevalence observed in female donors (female = 52/74 versus male = 33/66; p value = 0.01605 by Fisher's test; Figure S2b). From these, the eight

most neutralizing sera were subjected to mutational antigenic profiling. In contrast to mouse sera, human sera had heterogeneous antigenic profiles, varying in both the number and location of capsid sites conferring escape (Figs. 3 and 4). Overall, two types of profiles were observed: narrow profiles, where escape was focused in one region of the capsid, with a single residue harboring >50% of the observed positive differential selection (Fig. 3 and Figure S2) and broad profiles, with a wider distribution of escape across different

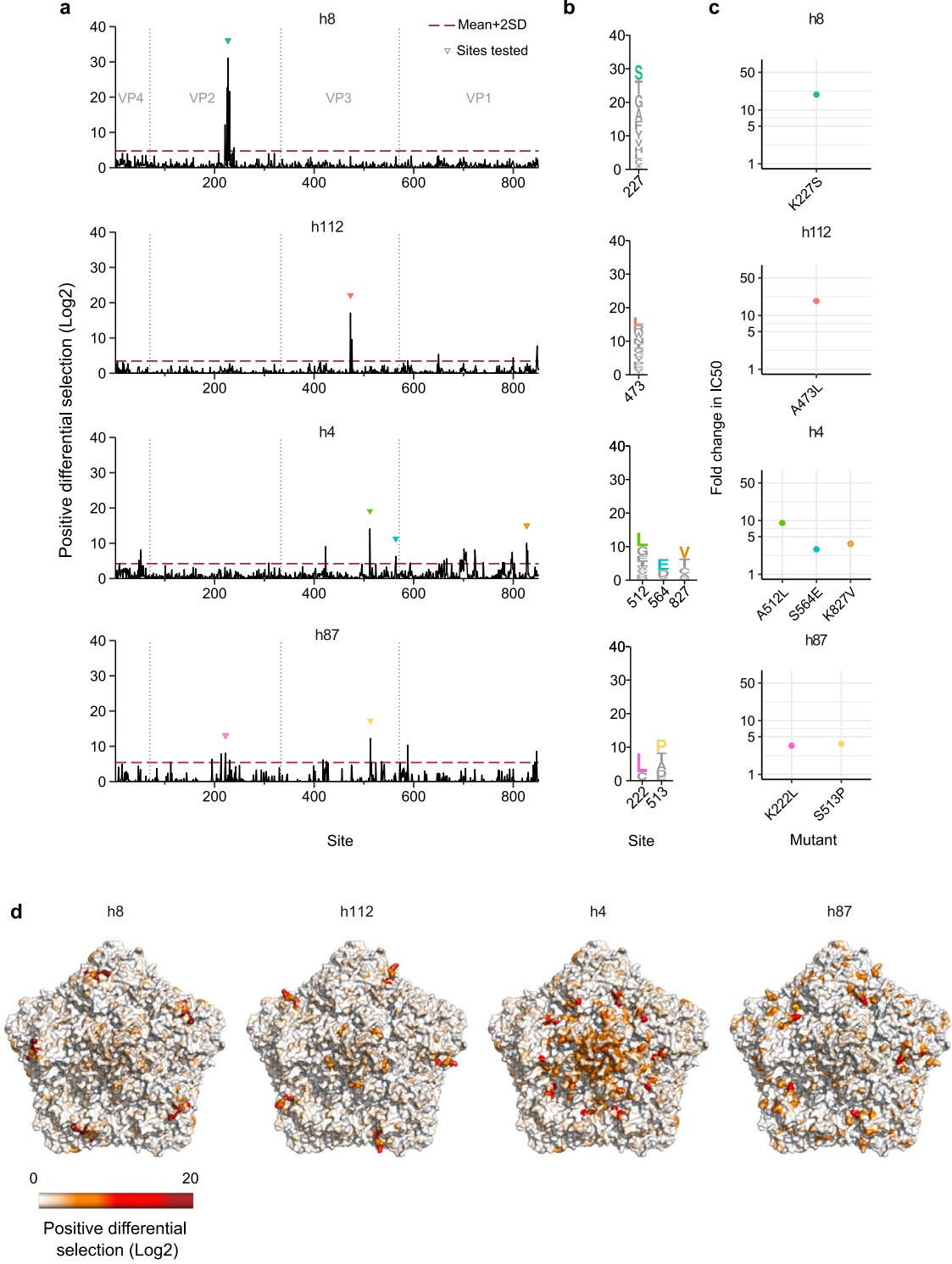

**Fig. 3 | Narrow antigenic profiles observed in four human sera. a** The antigenic profile of four human sera showing narrow escape profiles. Triangles indicate sites validated in **c. b** Logo plot representation of selected sites of escape evaluated in **c. c** The fold change in IC50 for the indicated mutants versus the WT virus. **d** The CVB3 capsid pentamer structure colored by positive differential selection values for each serum. Source data are provided as a Source Data file. See Figure S2 for the CVB3 neutralization titers in the evaluated donor sera, Figure S3 for individual replicates and the correlation between replicates, Table S1 for the surviving fraction of the viral populations after neutralization and Table S3 for the structural location of the main escape sites. See capsid_roadmaps for capsid roadmaps labeled by residue and dms_view for interactive visualization on GitHub[23].

capsid regions and no residue contributing >10% of the total positive differential selection (Fig. 4 and Figure S2c).

The narrow profile was observed in four of the eight human sera samples evaluated (Fig. 3 and Fig. S3). Escape mutations in these sera clustered into one or two of the four structural regions known to harbor antibody binding sites[17] (Fig. 3d). The most neutralizing serum (h8; reciprocal IC50 = 9186) showed escape in a single region of the VP2 EF loop (the "puff" region) in the outer surface of the canyon (Table S3) with multiple sites surrounding the main K227 escape residue involved (Fig. 3a, d). The strongest escape mutation, K227S,

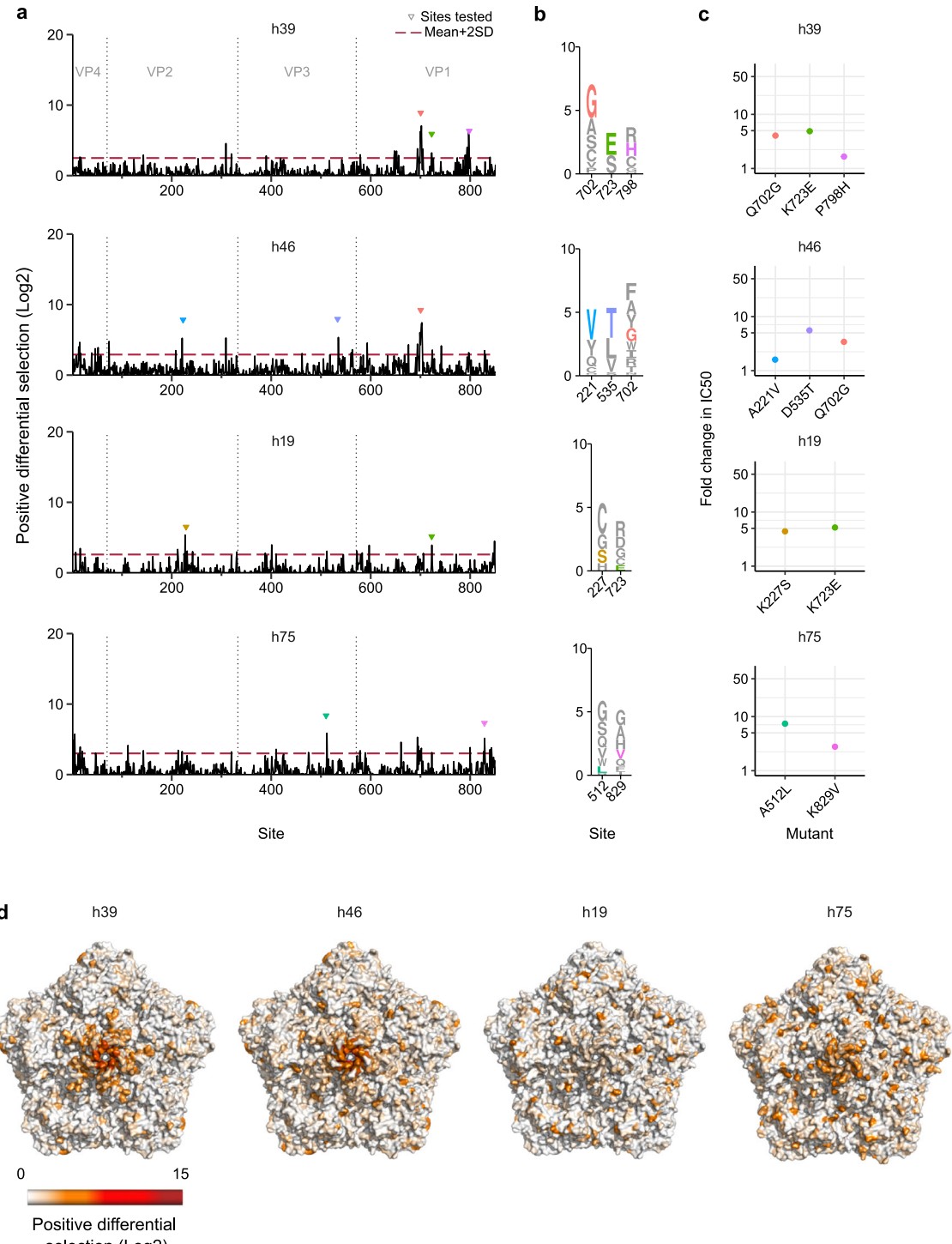

**Fig. 4 | Broad-antigenic profiles of four human sera. a** The antigenic profile of four human sera showing broad-antigenic profiles. Triangles indicate sites validated in **c. b** Logo plot representation of escape sites selected for validation in **c. c** The fold-change in the IC50 for the indicated mutations versus the WT virus. **d** The CVB3 capsid pentamer structure colored by positive differential selection values of the indicated serum. Source data are provided as a Source Data file. See

Figure S4 for individual replicates and the correlation between replicates, Table S1 for the surviving fraction of the viral populations after neutralization, and Table S2 for the structural location of the main escape sites. See capsid_roadmaps for capsid roadmaps labeled by residue and dms_view for interactive visualization on GitHub[23].

reduced neutralization by -19-fold (Fig. 3b, c). Interestingly, the main site of escape in this serum (K227) is the same as that conferring the strongest escape from the mAb yet the antigenic contexts involve different additional residues (see Figs. 1b and 3a). As for serum h8, serum h112 (reciprocal IC50 = 2359) presented a single prominent escape site in the threefold plateau (site 473, VP3 EF loop, Fig. 3a, d and

Figure S3). As anticipated, mutation of this site (A473L) also resulted in a strong escape (-19-fold change in the IC50).

Unlike sera h8 and h112, the antigenic profile of serum h4 (reciprocal IC50 = 3348) presented escape mutations across two capsid regions: the canyon inner surface (VP1 CD loop and VP3 CD and C-terminal loops) and the northern rim (VP1 DE, EF, and HI loops;

Fig. 3a, d and Figure S3). These sites are characteristic of human and murine antibodies that bind to the inner surface of the canyon in other enteroviruses[32,33]. Mutations in these three dominant sites resulted in IC50 fold changes of 8.99 (A512L), 3.69 (K827V), and 2.91 (S564E, Fig. 3c), in agreement with the antigenic profile. Finally, escape mutations in sample h87 (reciprocal IC50 = 3108) largely mapped to the interface between the inner surface (G2H loop of VP3) and the outer surface of the canyon (C-terminal loop of VP1 and EF loop of VP2; Fig. 3a, d and Figure S3). In agreement with the antigenic profile, more modest escape was observed for single mutations in each region, with K222L (EF loop of VP2) and S513P (G2H loop of VP3) resulting in a fold change in the IC50 of 3.42 and 4.71, respectively (Fig. 3c). Overall, the neutralization profiles defined above showed prominent sites of escape in one or a few contiguous regions of the capsid, with individual escape mutations resulting in reduced neutralization of 3–20-fold change.

The remaining four human sera presented a distinct antigenic profile, with escape sites distributing across multiple antibody neutralization regions with weaker overall differential selection (Fig. 4 and Figure S4; note the change in scale versus Fig. 3). In particular, sera h39 and h46 had similar neutralization capacities (reciprocal IC50 = 2520 and 2212, respectively) and similar mutational antigenic profiles, with the principal escape sites in the northern rim of the canyon. Indeed, escape from both sera was conferred by the same Q702G mutation within the shared region of VP1 (IC50 fold-change of 4.05 and 3.41, for serum h39 and h46, respectively; Fig. 4c). However, less-dominant sites varied between samples h39 and h46, targeting different regions of the outer surface of the canyon and the threefold plateau (Fig. 4a, d and Supplementary Table S3). Experimental evaluation of prominent escape mutations was in agreement with the antigenic profiles, with mutations in VP1 conferring escape from serum h39 (K723E and P798H; fold change in IC50 of 4.88 and 1.66, respectively) and mutations in VP2 and VP3 conferring escape from serum h46 (A221V and D535T; fold change in IC50 of 1.6 and 5.56, respectively).

Escape mutations in serum h19 (reciprocal IC50 = 2052) were broadly distributed across the capsid. Some of the main escape sites mapped to the VP2 EF loop and the VP1 C-terminal loop in the canyon outer surface, which are secondary structure elements also involved in the escape profile of serum h75 (reciprocal IC50 = 1414). In agreement with the profile of serum h19, mutation K227S (VP2 EF loop) and K723E (northern rim of the canyon) conferred fold changes in the IC50 of 4.4 and 5.19, respectively (Fig. 4c). The h75 profile presented additional sites of escape in the inner surface of the canyon (G2H loop of VP3) and the northern rim (DE loop of VP1). As expected, mutation A512L (inner surface of the canyon) and K829V (outer surface of the canyon) conferred escape from neutralization (IC50 fold-change of 7.41 and 2.78, respectively; Fig. 4c). Overall, escape mutations in broad-antigenic profiles were distributed across the capsid and conferred smaller changes in escape from neutralization (2–8-fold change in the IC50) in the majority of samples compared to narrow antigenic profiles.

### Escape mutations can be predicted based on structural and evolutionary parameters

Having obtained a comprehensive dataset of mutations conferring escape from eight different human sera, we next examined whether these mutations can be predicted based on different attributes using a machine-learning approach. To this end, we compiled for all possible mutations in surface-exposed capsid residues ($n = 5776$) a dataset comprising structural, physicochemical, and evolutionary parameters. In addition, we predicted the effects of all such mutations on protein stability, aggregation propensity, and disorder (19 numerical and categorical features in total; see analysis_sites_muts on GitHub[23]). This data was combined with a Boolean feature indicating whether the mutation led to strong escape in the antigenic profiling experiments ("escape") or not ("no escape"). Specifically, we define escape

mutations ($n = 213$) as those conferring strong escape on their own (mutational positive differential selection >2-fold) and occurring in a residue with a strong overall contribution to escape (site differential selection of >mean+2 SD in their respective profile).

To implement the machine-learning approaches, numerical features were first preprocessed, and the dataset was split into training and testing datasets, comprising 75% and 25% of the data, respectively (Fig. 5a, see Methods for details). Different algorithms were then evaluated. Overall, the random forest algorithm (RF)[34] using class priors and probability recalibration produced the best results, as measured by the $F_1$-score, accuracy, precision, and receiver operating characteristic (ROC) output (Table S4). When tested against the training dataset, the RF classifier showed 99.77 ± 0.05% accuracy (classification precision of 100% and 99.54 ± 0.10% for predicting no escape and escape, respectively; Fig. 5b and Table S5). However, while the accuracy was maintained at high values with the testing dataset (95.7 ± 0.5%) a significant drop in predictability for mutations conferring escape was observed (classification precision: no escape 97.0 ± 0.5%; escape: 37 ± 9%; Fig. 5b and Table S5).

The reduced predictability observed in the testing dataset for mutations conferring escape could stem from their low number in the dataset (~4% or $n = 53$) but could also result from incomplete identification of some mutations conferring escape in the antigenic profiling experiments. To address this possibility, we reran the RF algorithm on 1000 different random samples of training and testing datasets and identified mutations that were consistently classified as conferring escape by the RF algorithm (>70% probability) but not in the antigenic profiling experiments (see analysis_sites_muts on GitHub[23]). In total, 27 such mutations were identified. Interestingly, all of these occurred in 12 sites where strong escape was already observed with other mutations in the antigenic profiling experiments, strongly suggesting these are likely to confer escape. Reclassification of these 27 instances in the dataset increased the predictability of the RF classifier by 53% in every testing dataset following training (accuracy 96.4 ± 0.5%, classification precision: no escape 97.7 ± 0.4%; escape: 58 ± 7%; Fig. 5b and Table S5).

Taking advantage of the fact that escape mutations could be identified by the RF algorithm, we evaluated which features were most predictive of escape using both accuracy loss after feature shuffling (ALAS)[35] and Shapley additive explanations (SHAP) values[36] (Fig. 5c, d). Of the top seven features, the most relevant for forecasting escape were the identity of the WT and mutant AA. We therefore examined whether mutations conferring escape were enriched in particular residues, chemical polarity, or charge that could be important for antibody binding. Indeed, sites contributing to escape (WT AA) were enriched in polar and positively charged residues, in particular K and T ($p < 0.05$ by Fisher's exact test; Fig. 5e and Figure S5). In contrast, a significant depletion of non-polar residues ($p < 0.05$ by Fisher's exact test), and leucine in particular ($p < 0.01$ by Fisher's exact test), was observed at escape sites (Fig. 5e). On the other hand, mutations conferring escape (mutant AA) were significantly enriched in specific AA (R, A, G, L, and T; >2-fold enrichment; $p < 0.05$ by Fisher's exact test) and depleted in others (K and N; >2-fold reduction; $p < 0.05$ by Fisher's exact test) but this was not generalizable to other AA with the same characteristics ($p > 0.05$ by Fisher's exact test; Fig. 5f and Figure S5). The other five variables which contributed >10% to model accuracy were quantitative measures of the effect of mutations on stability ($\Delta\Delta G$), distance from the center, flexibility (B-factor value), predicted disorder, and natural variation (Shannon's entropy; Fig. 5c). Specifically, the combination of thermodynamically stable mutations in flexible, disordered, and distal surface residues that are evolutionarily variable were highly predictive of escape (Fig. 5c, d).

To validate the relevance of the top seven features identified above for predicting escape, we trained and tested an RF classifier using only these features on the dataset including the 27 reclassified

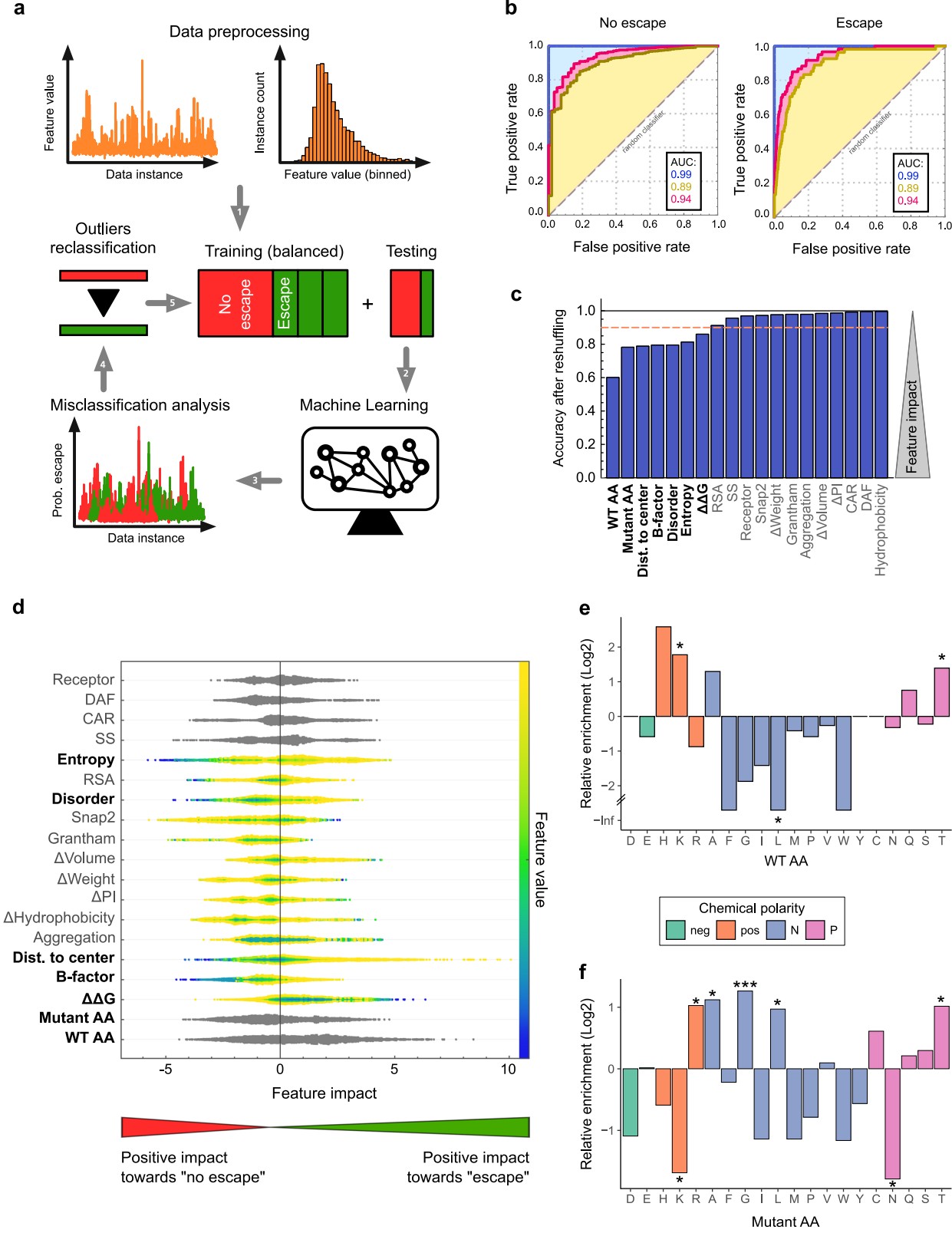

instances from above. Notably, the predictability of the RF classifier increased appreciably when evaluated on the testing dataset using only the top seven features versus the full set of features (accuracy 96.8 ± 0.5%, classification precision: no escape 97.4 ± 0.4%; escape: 72 ± 8%; Table S5). Together, these findings reveal mutations conferring escape can be predicted using machine-learning approaches and highlight the most relevant features for such predictions.

## Immune dominance of escape sites in the population

To better understand where polyclonal sera target the CVB3 capsid, we examined the distribution of sites conferring escape in the eight human antigenic profiles across the different capsid regions. Overall, 76 of 304 surface-exposed residues were observed to confer escape (25%). Of these, 39.5% (30/76) were located on the northern rim of the canyon, 34.2% (26/76) on the outer surface of the canyon, 17.1% (13/76)

**Fig. 5 | Escape mutations can be predicted by machine-learning approaches.**
**a** Overview of the machine-learning (ML) workflow: the dataset was preprocessed to reduce dimensionality by binning and randomly split into training (75%) and testing (25%) datasets. After 1,000 training-testing rounds, 27 outliers for which the ML classifier provided a large probability of escape (>70%) were identified. The reliability of the ML classifier was reevaluated after their reclassification (no escape to escape). **b** Receiver Operating Characteristic (ROC) curves for "no escape" (left) and "escape" (right) for training (blue), testing without reclassification (yellow), and testing with reclassification (pink). The area under the curve (AUC) is indicated in the insets. **c** Analysis of accuracy loss after feature shuffling. Features contributing to more than 10% of the model's accuracy are indicated in bold. **d** SHAP feature analysis. The colors indicate the relative value of numerical features and the positive or negative impact of the feature in the class is measured on the horizontal axis. Relevant predictive features (in bold) are those having a large impact, with their numeric value having directionality. **e** The relative enrichment of individual amino acids in sites conferring escape versus surface-exposed residues where no escape is observed. **f** The relative enrichment of individual amino acids in mutations conferring escape versus mutations in the same residues that do not confer escape. neg: negatively charged; pos: positively charged; N non-polar, P polar. *$p < 0.05$, **$p < 0.01$, ***$p < 0.005$ by a two-sided Fisher's exact test. Source data are provided as a Source Data file. See Figure S5 for characteristics of sites and mutations of escape and Tables S3 and S4 for a comparison of the different machine-learning algorithms and RF classifiers tested, respectively.

on the inner surface of the canyon, and 9.2% (7/76) in the three-fold plateau (Fig. 6a and Figure S6). Most samples present escape sites in the outer and northern rim of the canyon (Fig. 6a) which belong to secondary structure elements shown to confer escape from neutralization in other enteroviruses[18] (see Table S6 for comparison to other enteroviruses). To assess if specific regions and/or sites were similarly targeted in a larger sample size, we screened representative mutants from each capsid region for their ability to confer escape from 18 additional human sera (reciprocal IC50s of 550–1100; Fig. 6). Of the eight mutants tested, D535T (threefold plateau) and K723E (northern rim of the canyon) were the most common mutations affecting neutralization, conferring a >2-fold reduction in sensitivity in all sera (Fig. 6c). Reduced neutralization was also frequent with mutation P798H in the northern rim of the canyon (escape from 5/18 samples) and A512L in the inner surface of the canyon (escape from 4/18 samples). In contrast, the remaining mutations conferred escape from only a few (Q702G and K227S) or no (A473L and K650Y) sera despite being located in similar regions of the capsid as the most prevalent escape mutations (three-fold plateau and northern rim of the canyon, Fig. 6). Overall, these results uncover two conserved sites of escape in the three-fold plateau and northern rim of the canyon in all sera samples, and highlight that escape is specific to a given residue rather than a region.

## Strong escape is conferred by multiple mutations but comes at a cost to fitness

When single mutations conferred escape, their effects on neutralization were relatively moderate (average IC50 fold-change of $5.1 \pm 4.0$). As viruses can accumulate multiple mutations during adaptation, we assessed the ability of viral mutants encoding two escape mutations to evade neutralization by the top 26 most neutralizing sera. Surprisingly, a double mutant encoding the two most prevalent escape mutations at the population level, D535T and K723E, did not reduce neutralization compared to the single mutations alone and even reduced the degree of escape in some cases (Fig. 6c, average fold-change in IC50 of: $3.8 \pm 1.9$). In contrast, combining the first and the third most prevalent escape mutations (K723E + P798H), both of which are located in the northern rim of the canyon, conferred a similar or stronger escape from neutralization than the individual mutations in all but a single serum (h19; Fig. 6c). Indeed, the average reduction in sensitivity across all sera was increased by nearly two-fold when combining these mutations (average IC50 fold-change of $11.2 \pm 8.2$, $6.24 \pm 4.22$, and $2.36 \pm 0.39$ for K723E + P798H, K723E, and P798H, respectively; $p = 0.01$ and $p = 1.08 \times 10^{-5}$ vs. K723E and P798H, respectively, by a two-tail $t$ test). Moreover, in nearly 20% of the sera (5/26 sera), the K723E + P798H double mutant reduced neutralization by >20-fold versus the WT virus. Such escape was not observed for any of the single mutants, suggesting strong escape from antibody neutralization requires the combination of at least two mutations.

The ability of a virus to successfully infect a host is dependent on viral fitness. As some of the mutations examined above conferred escape from nearly all sera, we sought to determine whether variation at these particular sites is not observed due to strong effects on viral fitness. We therefore directly assayed the effect of all single and double escape mutations on viral fitness in the absence of antibody selection. Of the eight single mutants tested, five incurred a fitness cost compared to the WT virus. In particular, the two mutations conferring escape in all sera (D535T and K723E), reduced viral fitness to 20% and 30% relative to the WT virus, respectively ($p = 0.01$ and $p = 0.003$ by two-tail t-test for D535T and K723E, respectively; Fig. 6c). Combining these mutations further reduced fitness to 10% relative to the WT virus, significantly lower than each mutant alone ($p$ value = 0.0007 and $p = 0.0021$ versus D535T and K723E by two-tail $t$ test, respectively). The fitness of the mutant showing the strongest escape (K723E + P798H) was 50% lower than the WT ($p = 0.011$ by two-tail $t$ test) but was not significantly different from the fitness of the individual mutants ($p$ value > 0.5 by two-tail $t$ test for both). Overall, mutants conferring escape from multiple sera (>20% of the sera tested) had strong fitness costs (average fitness of $0.28 \pm 0.11$), while those conferring escape from only a few sera (<20% of the sera tested) were less deleterious (average fitness of $0.73 \pm 0.35$), although this difference did not reach statistical significance ($p = 0.08$ by a two-tail t-test). Together, these results suggest that escape from existing immunity may come at a significant cost to viral fitness, in particular for mutations in the most common sites.

## Discussion

Neutralizing antibodies play fundamental roles in resolving viral infections and provide long-term protection against reinfection. However, our understanding of how neutralizing antibodies in polyclonal sera target non-enveloped capsids has only recently begun to be addressed and remains relatively limited[37,38]. Herein, we map the neutralizing antibody responses of both humans and mice to a prototypical picornavirus. Analysis of polyclonal sera from infected mice revealed a homogenous antigenic profile (Fig. 2); indeed, all targeted a single discrete region of the capsid at the northern rim of the canyon, which was previously shown to be a target of neutralizing mouse mAbs[27–29]. In contrast to mice, the antigenic profiles from human sera were heterogeneous, targeting epitopes across all major antigenic regions in picornavirus capsids (Figs. 3 and 4). Moreover, two distinct types of antigenic profiles were observed. Four of the sera had a narrow antigenic profile, with neutralizing responses focused on one dominant antigenic site that, unlike the mouse sera, varied between individuals (Fig. 3). In contrast, the remaining four sera had broader antigenic profiles, with multiple regions being targeted co-dominantly (Fig. 4). Such responses are similar to those observed for the non-enveloped norovirus, where neutralizing polyclonal antibody responses target more than one immunodominant epitope and required multiple mutations for escape[38]. The different profiles observed in our study appear to have functional consequences: narrow profiles show an overall stronger neutralization capacity, comprising 4 of the 5 most neutralizing sera (labeled in bold in Fig. 6). Moreover, some of these sera (i.e., h8 and h112) are

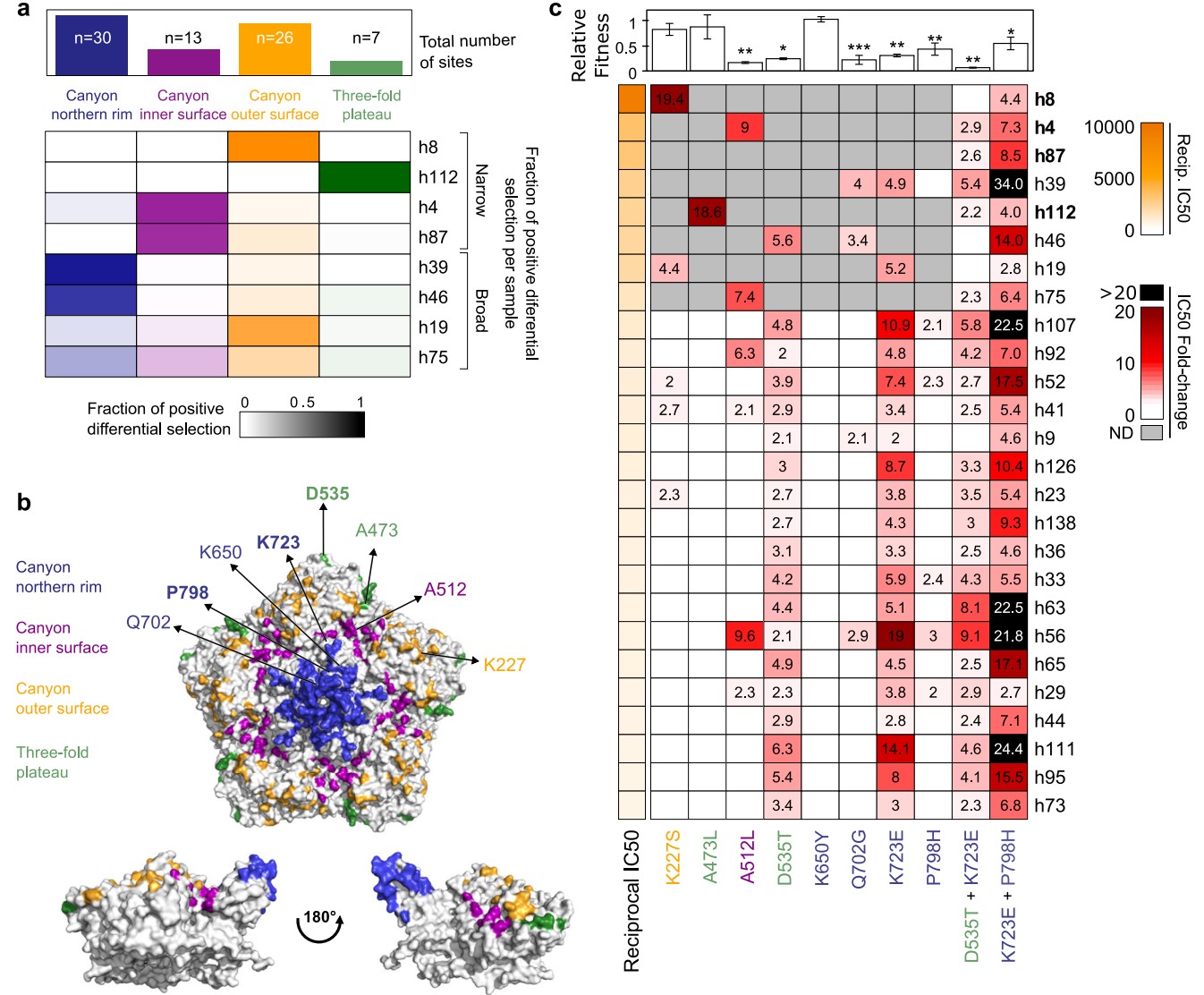

**Fig. 6 | Epitope dominance for neutralizing antibodies targeting CVB3.**
**a** Heatmap representation of the total (top) or per serum (bottom) positive differential selection by capsid region. **b** The distribution of the main escape sites identified in the human antigenic profiles mapped onto the CVB3 capsid pentamer, colored according to their structural location. The mutations evaluated **c** are indicated, with sites combined in the double mutants represented in bold and underlined. **c** Heatmap representation of the change in neutralization sensitivity conferred by single and double mutations. Samples with narrow antigenic profiles are highlighted in bold. Numbers indicate fold change in the IC50 of the mutant versus the WT virus when exceeding twofold. ND not determined. The upper bar plot indicates the relative fitness of each mutant versus the WT virus. \*$p < 0.05$, \*\*$p < 0.01$, \*\*\*$p < 0.005$ by two-tailed $t$ test. Source data are provided as a Source Data file. See Figure S6 for a heatmap representation of positive differential selection per secondary structure element in the capsid.

more susceptible to immune evasion, with single mutations conferring a stronger degree of escape than in broad-antigenic profiles (Figs. 3 and 4).

Analyses of polyclonal immune responses to the membrane proteins of enveloped viruses have also revealed both narrow and broad neutralization profiles. Narrow profiles were reported for neutralizing antibodies targeting the influenza A hemagglutinin (HA) protein, which were focused on a single region and could be efficiently evaded by individual mutations[9]. In contrast, polyclonal responses to the H glycoprotein of measles virus were shown to co-dominantly target multiple sites[39], with individual mutations conferring weak escape from neutralization. Interestingly, antibodies targeting influenza A virus generally do not confer significant protection against reinfection with related strains, while those targeting the measles virus H protein do[40], raising the possibility that individuals with broad-antigenic profiles are more resistant to reinfection with related viruses. It is currently unclear what drives the evolution of broad or narrow antigenic profiles. One possibility is that broad profiles represent the consequence of multiple CVB3 infections. Since neutralizing antibody epitopes show significant variability (Fig. S5d), repeated exposure to different CVB3 variants may broaden immune responses, as has been suggested for norovirus infections[41]. Unfortunately, we are unable to address this question in the current study as the number of times the donors in our study were infected with CVB3 and the infecting strain(s) are unknown. In this sense, analysis of the antigenic profiles from children may be informative to distinguish these possibilities as, unlike adults, they are likely to have been exposed to a single CVB3 variant. Finally, cross-neutralization by different picornaviruses has been recently demonstrated in mice[42], and could also influence the breadth of neutralizing antibody responses. However, this does not seem to underlie the difference between sera showing narrow or broad neutralization profiles as a preliminary analysis showed these to have similar neutralization capacity against five different CVB clinical strains (CVB1, 3, 4, 5, and 6; Fig. S7).

Having obtained a comprehensive dataset of mutations conferring escape in eight human sera, we sought to understand whether such mutations could be predicted using machine-learning approaches and to define the most relevant attributes for such prediction. We found that an RF could predict escape to relatively high accuracy based on only seven features, indicating that mutations conferring escape have unique, typifying attributes (Fig. 5). Specifically, we found escape mutants to be enriched in particular amino acid combinations, to be less destabilizing, and to occur in residues that are flexible, distal from the core, and evolutionarily variable (Fig. 5). As these parameters can be obtained for any virus for which a high-resolution capsid structure is available, this machine-learning algorithm could potentially be used to identify escape mutations in related picornaviruses without the need to perform antigenic profiling.

The relative dominance of epitopes targeted by polyclonal human responses following natural infection with CVB3 is unknown. To address this, we assessed the ability of individual escape mutants from different capsid regions to alter neutralization by an additional 18 sera (Fig. 6). Overall, most human sera targeted sites around the canyon (Fig. 6a), as has been observed for enterovirus A71[43]. However, several additional observations were made from this analysis. First, the two mutants conferring the most significant escape in the antigenic profiles, K227S and A473L (>18-fold reduction in IC50; Fig. 6), which were derived from sera showing narrow antigenic profiles, did not confer significant escape in the remaining sera (maximum reduction in IC50 of 4.4-fold). Second, the most frequent mutations conferring escape were observed in broad-antigenic profiles. Specifically, the D535T (3-fold plateau) and K723E (northern rim) mutations conferred some degree of escape in all sera tested, highlighting a conserved neutralizing antibody response in human polyclonal sera to natural CVB3 infection. Structural analyses of murine polyclonal antibodies targeting CVA21 have also shown these two capsid regions to harbor immunodominant epitopes[44], suggesting this conservation may extend to additional enteroviruses. Finally, the K650Y mutation that conferred strong escape from all mouse sera had no appreciable effect in any of the human sera, despite belonging to the canyon northern rim epitope encompassing residue 723, which was targeted by all human sera (Figs. 2 and 6c). This is likely due to sequence differences, with the strain used for the mouse experiment encoding a lysine at residue 650, while the majority of available CVB3 sequences have a glutamic acid at this position (~95%; Supplementary Data file 1).

Having established the epitope dominance of individual escape mutants, we next assessed the degree of escape conferred by combining two individual escape mutations. Somewhat unexpectedly, a virus encoding the two mutations conferring reduced neutralization by all sera tested (D535T and K723E) did not lead to increased escape compared to the individual mutations (Fig. 6c). In contrast, combining the two most frequent mutations in the northern rim of the canyon (K723E and P798H) resulted in enhanced escape from multiple sera (20 to 34-fold reduction in IC50; Fig. 6c). As homologous residues in CVA6 and Echovirus 30 were shown to be bound by mAbs[29,45], these likely comprise a single epitope in CVB3 that requires two mutations to be efficiently escaped. While this could suggest this epitope to be an attractive target for vaccines, both residue 723 and 798, as well as the adjacent residues are variable in nature (see table CVB3_nancy_table_escape_muts_sites on GitHub[23]) and mutation of both residues showed similar fitness to that of the individual mutants, suggesting escape could rapidly arise (Fig. 6c).

Overall, our study highlights the utility of mutational antigenic profiling for understanding neutralizing antibody responses to non-enveloped viruses. The application of this technique to additional viruses of this group can help shed light on variation in polyclonal responses across different viruses and may provide clues as to why some viruses harbor only a limited number of serotypes (e.g. three for poliovirus) while others can reach much higher antigenic diversity (>100 for rhinovirus[13]). Nevertheless, it is important to note that the methodology used in this study can only detect the effects of single mutations. Indeed, only by combining two of the mutations identified in the different profiles were we able to show the prevalent and, at times, dominant escape conferred by combined mutations at sites 723 and 728 (Fig. 6c). Moreover, the detection of subdominant epitopes in our assay may also be limited, as escape at these sites may be masked by stronger selection at more dominant epitopes[46]. Recently, mutagenesis protocols that introduce a small fraction of all possible double mutations at specific, known antigenic sites and employ long-read sequencing technologies that allow for haplotype assignation have been used to overcome some of these limitations[46–48]. The information obtained in the current study can inform the selection of relevant mutations for assessing the impact of double mutations on escape from polyclonal sera. Alternatively, experimental evolution of viruses harboring mutations conferring escape in dominant sites in the presence of neutralizing sera could be used to uncover subdominant epitopes.

## Methods

The performed study complies with ethical regulations. For human sera samples, approval was obtained from the IBSP-CV Biobank (PT17/0015/0017), integrated in the Spanish National Biobanks Network and in the Valencian Biobanking Network. Animal work was approved by the Valencian government's ethics committee (approval 2019/VSC/PEA/0151). Experiments with infectious viruses and genetically modified organisms were approved by the Spanish committee on genetically modified organisms, the University of Valencia's biosafety committee, and the institute's biosafety committee.

### Cells, viruses, and reagents

HeLa H1 (CRL-1958; RRID: CVCL_3334) and HEK293T cells (CRL-3216) were obtained from ATCC and were periodically validated to be free of mycoplasma. Cells were cultured in culture media (Dulbecco's modified Eagle's medium, Pen-Strep, and L-glutamine) supplemented with 10% or 2% heat-inactivated fetal bovine serum for culturing or infection, respectively. The anti-Coxsackievirus B3 monoclonal antibody was obtained from Merck (clone 280-5F-4E-5E, MAB948). The human codon-optimized T7 polymerase plasmid was obtained from Addgene (#65974) and the CVB3 infectious clones encoding mCherry (CVB3-mCherry) or eGFP (CVB3-eGFP) were previously described[49]. The titer of these reporter viruses was obtained by infecting Hela H1 cells with serial dilutions of each virus in 96-well plates and counting the number of fluorescent cells at 8 h post-infection using an Incucyte SX5 Live-Cell Analysis System (Sartorius). The CVB3 populations harboring high diversity in the capsid region have been previously described[22]. Briefly, a reverse genetics plasmid encoding CVB3 genome was mutagenized at the codon level to generate viral populations containing >96% of all possible single amino acid mutations in the capsid region. The virus populations were then passaged at low multiplicity of infection (MOI) to create a passage 1 population in which there is a genotype-phenotype link between the capsid proteins of each virus and the genome it carries. The viral titers in these libraries were determined by plaque assay in Hela H1 cells. The coxsackievirus clinical strains were obtained from EVAg and were: CVB1 (EVAg RO-98-1-74; GenBank LS451286), CVB3 (EVAg RO-123-1-95; GenBank LS451287.1), CVB4 (EVAg RO-69-1–86; GenBank LS451289), CVB5 (EVAg RO-14-5-70; GenBank LS451290), and CVB6 (EVAg RO-86-1-73; GenBank LS451291)[50]. These viruses were treated with chloroform to remove mycoplasma, amplified in HeLa H1 cells for 2 passages, and titered by limiting dilution at 7 days post-infection. All virus experiments were carried out under BSL2 conditions after obtaining approval from the biosafety committees of both I2SysBio and the University of Valencia. All work with genetically modified organisms was approved by the relevant national committees.

## Neutralizing sera

The 140 human sera used in this study were provided by the IBSP-CV Biobank (PT17/0015/0017), integrated into the Spanish National Biobanks Network and in the Valencian Biobanking Network and they were processed following standard operating procedures with the appropriate approval of the Ethics and Scientific Committees and with signed informed consent from the participants. All samples were collected from healthy donors between 2010-2021 and included 74 females and 66 males. No information was available on the CVB3 infection record of these individuals. For mouse sera (ethics protocol approval 2019/VSC/PEA/0151), five-week-old male Balb/C mice (Charles River, strain code 028) were infected via intraperitoneal injection with $10^4$ (m1) or $10^5$ (m2 and m3) PFUs of CVB3 and sera were collected three weeks post-infection. All sera were heat-inactivated for 30 min at 55 °C before use.

## Antigenic profiling

For mutational antigenic profiling, we used the passage 1 mutant virus populations derived from libraries 1 and 2 (L1 and L2) described in[22]. For each serum or antibody, we performed two biological replicates (i.e., L1 and L2), with one or two replicates of each of the virus libraries (technical replicates). Briefly, $10^6$ PFU of each virus library were mixed with sera or antibodies at the indicated dilutions to have 1% to 10% of the virus library survive the neutralization treatment, as judged by qPCR (see below). Mock-neutralized controls were included for each neutralized library replicate. After a 1 h incubation at 37 °C, Hela H1 cells plated the day before in six-well plates were infected with the virus-antibody mix or virus alone. At 8hpi, cells were subjected to three freeze–thaw cycles, cell debris was removed by centrifugation at $500 \times g$, and the supernatants were collected. Finally, 200 µL of each sample were treated with 2 µL of RNase-Free DNaseI (ThermoFisher) for 30 min at 37 °C, and viral RNA was extracted using the Quick-RNA Viral Kit (Zymo Research), eluting in 20 µL.

To estimate the fraction of virus surviving the neutralization treatment, viral RNA was quantified by RT-qPCR using the GoTaq Probe 1-Step RT-qPCR System (Promega) on a QuantStudio3 (ThermoFisher Scientific). Ten-fold serial dilutions of RNA extracted from a known titer virus stock were used to generate a standard curve. RT-qPCR was performed using 2 µL of template RNA of neutralized and control samples (forward primer: RTqPCR_F (GAT CGC ATA TGG TGA TGA TGT GA), reverse primer: RTqPCR_R (AGC TTC AGC GAG TAA AGA TGC A), and TaqManProbe: CVB3_probe (6FAM-CGC ATC GTA CCC ATG G-TAMRA). RT-qPCR Ct values were interpolated to the standard curve to quantify viral RNA in each sample and the fraction of virus surviving neutralization was determined versus the untreated controls (see Table S1 for details on the number of replicates performed, amount of antibody used, and surviving fraction after neutralization).

## Duplex sequencing

For sequencing, viral RNA extracted as indicated above was reverse transcribed with the high-fidelity OneScript Plus Reverse Transcriptase (Applied Biological Materials) using 8 µL of RNA and the primer CVB3_RT_3450 (GTGCTGTGGTCGTGCTCACTAA). To ensure sufficient diversity was present in the cDNA, the number of capsid copies in the cDNA was quantified via qPCR using PowerUp SYBR Green Master Mix (ThermoFisher Scientific) in a 10 µL total reaction, using 1 µL of the template and the primers CVB3_qPCR_F (CCCTGAATGCGGCTAATCC) and CVB3_qPCR_R (AAACACGGACACCCAAAGTAGTC). A standard curve was generated using the original reverse genetic plasmid encoding the CVB3 genome. The full capsid region was then amplified from the cDNA using KOD polymerase Mastermix (Novagen) and the primers CVB3_P1_seq_F (CCCTTTGTTGGGTTTATACCACTTAG) and CVB3_P1_seq_R (CCTGTAGTTCCCCACATACACTG). Duplex sequencing libraries were prepared as previously described[22] and sequenced on an Illumina Novaseq 6000 sequencer. The resulting files were

analyzed as previously described[22] to obtain the counts of each codon at each position (codon tables, available on GitHub[23], section 1). As before, all single mutations in codons were omitted from the analysis to increase the signal-to-noise ratio[22].

## Calculation and visualization of positive differential selection

Codon tables were analyzed for differential selection versus mock-neutralized controls of each library using the DMS_tools2 package[51] with the default settings. Two batches of experiments were performed with a different number of replicates. The first batch included samples mAb, m1, m2, m3, h4, h8, h39, and h46, and three non-neutralized controls for each library. Four replicates were performed for each sample (2 technical replicates of 2 biological replicates from libraries 1 and 2). Differential selection of each sample was calculated versus the mean of the three controls. In the second batch, 2 biological replicates (libraries 1 and 2) were performed for samples h19, h75, h87, and h112 together with an individual non-neutralized control for each library. One of the replicates of h87 (library 1) had to be discarded due to the low mutation rate observed in this sample after sequencing (30% smaller than the average of the other selected libraries), and positive differential selection was calculated versus the individual control of each library. All positive differential selection plots in the main figures represent the mean differential selection for each mutation at each particular site across replicates. Line plots represent the mean positive differential selection per site and logoplots represent the mean positive differential selection for each amino acid mutation at selected sites. Differential selection tables and positive differential selection logoplots of the full capsid region are available on GitHub[23] (sections 2–4). Interactive visualization of the antigenic profiles using the dms-view software[52] is available on GitHub[23] (section 6).

## Neutralization assays

For fluorescence-based neutralization assays, $2.5 \times 10^3$ fluorescent virus units of WT or mutant CVB3 mCherry virus were mixed in duplicate with serial dilutions of each antibody or sera sample and incubated for 1 h at 37 °C. Virus-only samples without antibodies were used as a control to measure maximal fluorescence in the absence of neutralization. Virus-antibody mixes were then transferred to 96-well plates containing Hela H1 cells plated the day before. After an 8-hour incubation at 37 °C, mCherry-derived red fluorescent signal was measured using an Incucyte SX5 Live-Cell Analysis System (Sartorious) to determine fluorescent virus units per well. The antibody concentration resulting in a 50% reduction of virus signal (IC50) was obtained using the two-parameter log-logistic function LL2 from the R drc package (version 3.0.1). All IC50 values represent the mean and standard error of the curve fitting of two replicate curves run in the same 96-well plate. Results of the human samples neutralization screening (section 7) and the neutralization assays performed with different CVB3 mutants (section 8) are available on GitHub[23].

For cytopathic effect (CPE)-based neutralization assays, two-fold serial diluted sera were mixed with an equal volume of virus containing 100 TCID50/well of the virus. The virus-antibody mix was incubated at 37 °C for 1 h and then transferred to 96-well plates containing Hela H1 cells plated the day before. After an incubation of 5 days, cells were fixed with 5% PFA and stained with crystal violet. The neutralization titer was determined as the reciprocal of the highest dilution at which at least 50% of wells showed complete inhibition of CPE. All neutralizations were tested in duplicates. Results of the neutralization assay of different CVB strains are available on GitHub[23] (section 8)

## Generation of CVB3 capsid mutants

CVB3 mutants were generated by site-directed mutagenesis using the mCherry-CVB3 fluorescent infectious clone mentioned above[49]. For each mutant, non-overlapping primers containing the desired mutation in one of the primers were used to introduce the mutation with Q5

polymerase, followed by DpnI (Thermo Scientific) treatment, phosphorylation, ligation, and transformation of chemically competent bacteria (NZY5α Competent Cells, NZY Tech). Successful mutagenesis was verified by Sanger sequencing. Subsequently, plasmids were linearized with SalI-HF (ThermoFisher), and 600 ng of plasmid were transfected into $5 \times 10^4$ HEK293T cells, together with 600 ng of a plasmid encoding the T7 polymerase (Addgene 65974) using Lipofectamine 2000 (ThermoFisher) according to manufacturer's instructions of use. Cells were then incubated until cytopathic effect (CPE) was observed and passage 0 virus was collected. Infectious virus titer was determined using an Incucyte system (Sartorius) to quantify the red fluorescence from the mCherry reporter signal as detailed above. If needed, mutants with low titers were amplified for an additional passage. The capsid region of the mutant virus populations was reverse transcribed, PCR-amplified, and sequenced to ensure no compensatory mutations or reversions arose during replication.

### Competition assays to assess the fitness of virus mutants

A 1:1 mixture of CVB3-eGFP and either CVB3-mCherry viruses encoding the WT or mutant capsid was used to infect Hela H1 cells at an MOI of 0.001 in a 24-well plate. Automated real-time quantitative fluorescence microscopy was used to track the spread of each virus in an Incucyte SX5 Live-Cell Analysis System (Sartorious). The ratio of the mCherry signal (derived from the WT or mutant virus) to the eGFP signal of the reference virus at 20hpi ($Ratio_{20hpi}$) provides a measure of the relative success of both the WT and mutant virus following several rounds of infection. The ratio of mCherry to eGFP signal at 8hpi ($Ratio_{8hpi}$) reflects the initial relative abundance of each virus before the competition. To calculate the fitness of each mutant versus that of the WT virus, the formula $(Ratio_{20hpi}^{Mutant}/Ratio_{8hpi}^{Mutant})/(Ratio_{20hpi}^{WT}/Ratio_{8hpi}^{WT})$ was used. The results of the competition assay are available on GitHub[23] (section 9).

### Bioinformatic analyses

For bioinformatics analysis, R version 4.2.0 was used with packages tidyverse, reshape2, drc (IC50 curve fitting), ggplot2 and ggh4x (plotting), ggseqlogo (logoplots), ComplexHeatmap (heatmaps), viridis (color scale), ggcorrplot (correlation matrices), igraph (bpartite network analysis), DECIPHER and Biostrings. R[53], PyMOL[54], and Inkscape[55] were used to generate the figures. CVB3 Nancy capsid roadmaps were generated using RIVEM[56] with surface-exposed residues colored according to their positive differential selection values and labeled according to the position of each residue in the polyprotein, available on GitHub[23], section 5.

All structural features were calculated based on a CVB3 capsid pentamer (PDB ID: 4GB3) mutated to encode the Nancy strain amino acid sequence with FoldX[57] as previously reported[22]. Outer surface residues were defined as residues with a distance to the center of the capsid >130 Å and a surface exposure >2.5 Å² based on the FindSurfaceResidues script in PyMol[54]. The relative surface area (RSA) of residues in the pentamer was calculated using PSAIA[58]. For entropy calculation, all available enterovirus A, B, and C full polyprotein sequences in BV-BRC (bv-brc.org) were downloaded on May 14th, 2022. Duplicate sequences were removed, and the alignments were randomly subsampled to harbor the same number of sequences using the Biostrings package in R. Sequences were then combined and aligned using the DECIPHER package in R. All gaps versus the CVB3 Nancy capsid sequence were removed, and Shannon's entropy was calculated using a custom R script. Both the alignment file and the Shannon entropy can be found on GitHub[23] (section 11). Finally, B factors were obtained from the PDB file, distance to center was calculated in PyMol using the command distance (cmd.dist),disorder was calculated on the capsid sequence using IUpred3[59] and Snap2 scores were obtained from SNAP2[60]. The ΔΔG grantham, aggregation, and aa properties were calculated as previously described[22]. Tables used for the analysis of the

characteristics of the main escape sites and mutations are available on GitHub[23] (section 10).

### Machine learning

The initial dataset used for the machine-learning analyses was obtained by merging the structural, physicochemical, and evolutionary features for all surface-exposed capsid residues (see Bioinformatic analyses section for details). Sites of escape were defined as surface-exposed residues with a positive differential selection value above the mean+2 SD value per sample. Within these sites, escape mutations for each sample were defined as mutations with a >2-fold positive differential selection value (see GitHub[23], section 10). To implement the analysis, the data was preprocessed. Specifically, some of the numerical features in the original dataset had nearly as many different values as the number of instances ($n = 5776$). This complexity (dimensionality) was reduced to increase the predictive capabilities of ML by implementing a binning approach that assumes an underlying smooth distribution of values and assigning all the instances falling within a bin the same value (the bin mid value). For example, ΔΔG, which had 5,772 values originally, was reduced to 93 values following binning.

Next, the dataset was randomly sampled and split into training (75%) and testing (25%) datasets for each iteration of training and testing. The training dataset was balanced in terms of "escape" and "no escape" entries by repeating the "escape" entries. Following training, we evaluated the performance of three algorithms: neural network, support vector machine, and random forest (Table S4). Given that our data was clearly biased towards the "no escape" class, we assessed the possible improvement when using prior probabilities and a probability recalibration approach. Overall, the RF combined with prior probabilities and probability recalibration showed the best performance.

Finally, to identify mutations that were consistently misclassified in the ML compared to the antigenic profiling dataset, we estimated the expected number of times that the same instance is wrongly classified in $N$ different testing samples of a dataset by a random classifier using the formula $Nf_{test}/2$, where $f_{test}$ is the fraction of data for testing. We then sampled $N = 1,000$ different training/testing sets and, for each sample, we identified during the testing step the wrongly classified instances (with respect to the antigenic profiling data) having a probability >70% to belong to the "escape" class. We then selected among the ones that appeared more than expected ($Nf_{test}/2 = 125$ times). This procedure yields the list of 27 instances that we reclassified (see GitHub[23], section 10). All machine-learning analyses were performed using Mathematica 13.0[61] and the code is available on GitHub[23] (section 12).

### Statistical analyses

Wilcoxon tests, Fisher's exact tests, and t-tests were performed in R[53]. ns: $p > 0.05$, *$p < 0.05$, **$p < 0.01$, ***$p < 0.005$, ****$p < 0.001$. All tests were two-tailed.

### Reporting summary

Further information on research design is available in the Nature Portfolio Reporting Summary linked to this article.

## Data availability

The sequencing data generated in this study have been deposited in the SRA database under the accession code: BioProject PRJNA779606, SRA: SRP345591. The following supplementary files are available on GitHub (doi.org/10.5281/zenodo.8278367)[23]: 1, **Codon_tables.** Codon files for all samples; 2, **Meansitediffsel.** Mean positive site differential selection per site calculated for each sample. A subfolder is included with the positive differential selection of each replicate; 3**, Meanmutdiffsel.** Mean positive site differential selection per mutation

calculated for each sample. A subfolder is included with the positive differential selection of each replicate; 4, **Logoplots**. Logoplots representing the mean positive differential selection values of positively selected mutations for each sample; 5, **Capsid_roadmaps**. Capsid roadmap with surface-exposed residues colored according to the positive differential selection values obtained and labeled according to their position in the polyprotein; 6, **Dms_view**. Files for the interactive visualization of the antigenic profiles in dms-view (dms-view.github.io).7, **Biobank_results**. Results of the neutralization screening performed for 140 human samples. Anonymized ID, sex, and age of donors are indicated, together with collection date, ic50 obtained, and sd error in ic50 calculation; 8, **Neutralization_results**. Results of the neutralization assays for the human and mouse sera with different CVB3 mutants and coxsackievirus B clinical strains. Contains the IC50 results of the neutralization assays for the human and mouse sera samples. Sample and virus tested are indicated, together with the ic50 obtained for the mutant and for the WT virus, and the fold change in the IC50. Also contains the neutralization titers of the different human sera samples for different coxsackievirus B clinical strains; 9, **Competition_assay**. Data from the competition assay. The ratio between fluorescence signal obtained at 20hpi versus 8hpi for each mutant or wildtype virus is indicated.10, **Analysis_sites_muts**. Tables were used for the structural analyses and prediction of characteristics of the main escape sites and mutations. Contains a table for converting polyprotein position to chain and residue of the capsid proteins; 11, **Sequence_alignments**. Alignment file of enterovirus A, B, and C full polyprotein sequences in BV-BRC (bv-brc.org) and the calculated Shannon entropy. Source data are provided with this paper.

## Code availability
The following custom code is available on GitHub (doi.org/10.5281/zenodo.8278367)[23], section 12, **ML_code**. Custom code used for the machine-learning analyses in Mathematica.

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

## Acknowledgements

We want to particularly acknowledge the donors of the sera samples and the IBSP-CV Biobank (PT17/0015/0017) integrated into the Spanish National Biobanks Network for their collaboration. We would like to thank Luciana Rusu and Joao Zulaica for their technical support and Drs. Rafael Sanjuan and Santiago F. Elena for helpful discussions. The CVB3 clinical strains have been received from Mariana Combiescu, Cantacuzino Institute, Bucharest, Romania. This publication was supported by the European Virus Archive goes Global (EVAg) project that has received funding from the European Union's Horizon 2020 research and innovation program under grant agreement No 653316. Finally, the authors would like to acknowledge the use of the Principe Felipe Research Center (CIPF) server which was co-financed by the European Union through the Operativa Program of the European Regional Development Fund (ERDF/FEDER) of the Comunitat Valenciana 2014–2020. This work was supported by grants to R.G. from the Spanish Ministerio de Ciencia y Innovacion (grants BFU2017-86094-R and PID2021-125063NB-I00), the Generalitat Valenciana (grant AICO/2020/216), and the European Commission NextGenerationEU fund through Consejo Superior de Investigaciones Científicas (CSIC's) Global Health Platform (Plataforma Temática Interdisciplinar, PTI Salud Global). B.A.-R. holds a Generalitat Valenciana postdoctoral fellowship (APOSTD/2021/017). J.B. acknowledges support from the Spanish Ministry of Science and Innovation through grants PID2022-137436NB-I00 and PID2019- 105566GB-I00 and from the LifeHUB Research Network through grant PIE-202120E047-Conexiones-Life (CSIC).

## Author contributions

B.A-R., J.B., and R.G. designed and analyzed the data and wrote the manuscript. R.G. and J.B. acquired funding. R.G. conceived the project. J.B. designed and implemented the machine-learning analysis. B.A-R. performed all experiments.

## Competing interests

The authors declare no competing interests
