## [Peer Review File · Nature Communications]

REVIEWER COMMENTS

Reviewer #1 (Remarks to the Author):

Review of "Comprehensive profiling of polyclonal sera targeting a non-enveloped viral capsid"

This paper uses deep mutational scanning to look at antibody escape mutations from monoclonal antibodies, polyclonal mouse serum, and polyclonal human serum. The experiments work effectively as validated by neutralization assays, and provide new insight into the antigenicity of the CVB3 capsid proteins. The authors also making interesting observations about the variability of specificity among humans (as compared to mice) and the breadth of epitopes targeted relative to other viruses.

Overall, this is an interesting and well done study, and I support its publication with minor comments.

MAJOR COMMENTS:

- It would be helpful to discuss more why human sera have different epitopes? The authors mention in discussion the possibility of variable exposure histories. More elaboration would be useful: for instance, how variable are circulating CVB3-like viruses at these sites?

- I found the machine learning section less compelling than some of the others. Wildtype amino acid identity helps predict which sites are escape sites, but is this just confounded with surface accessibility: is it surface accessible sites that mediate escape and those tend to be charged / polar? Beyond that, what are we learning from machine learning? If different human sera have different escape mutations then clearly that must be driven by some non-viral factors (that aren't included in the machine learning model).

- Since these experiments create mutants of a human pathogenic virus, some discussion should be given to biosafety implications of making these mutants. Do the experiments combining mutations create mutants with more antigenic escap

e than likely to arise naturally? How did the authors think about this, and what types of biosafety considerations s

ould apply to this and similar research?

MINOR COMMENTS:

- Second paragraph of Results, the text refers to an interactive version of Figure 1F and links to the GitHub. However, when I navigate to the GitHub it is not really clearly documented how to find this particular interactive visualization (I couldn't find it). Maybe a more explicit link could be provided, or more details in the GitHub README? The same point goes for other links to the GitHub throughout the document: it's hard to figure out exactly where in the GitHub the relevant data are found. All of this could probably be fixed with a better README.

- Figure 2: the three mice all share the same y-axis scale. Maybe this is reasonable to do, but I would not object if the scales were different for the mice either since often the extent of differential selection in such experiments depends on the serum concentration and IC50 which may differ a bit. Basically, it's fine as is, but each could also be on its own scale. Similar comments apply to some other figures.

- Is there a biological explanation why we might expect neutralization to be different between male and female humans?

- In Figure S3 and S4, why did replicates per sera differ between individual sera? This is probably fine and I am not asking for more replicates, but just explanation of this fact would be helpful.

Reviewer #2 (Remarks to the Author):

Alvarez-Rodriguez et al., obtained the full antigenic profile of multiple human and mouse sera targeting the capsid of a prototypical picornavirus. Their results uncover significant variation in the breadth and strength of neutralization sites targeted by individual human polyclonal responses, which contrasted with homogenous responses observed in experimentally infected mice. Also, they use these comprehensive antigenic profiles to define key structural and evolutionary parameters that are predictive of escape, assess epitope dominance at the population level, and reveal a need for at least two mutations to achieve significant escape from multiple sera. Overall, data provide the first comprehensive analysis of how polyclonal sera target a non-enveloped viral capsid and help define both immune dominance and escape at the population level in tissue culture.

Minor:

All figures:

Why did the authors use the structure of strain SD (PDB ID:4GB3)?

Figure 4c comment:

Please correct the order of mutations for sera h39.

Figure 6c comment:

Please include the ND in Figure 6C.

Page 7 - "Unexpectedly, all three sera showed a similar antigenic profile, suggesting that the response of experimentally immunized mice is relatively invariant..." - Why is this result unexpected given the genetic inbreeding of laboratory strain mice?

REFERENCES

References 15 and 31 are the same one.

Major:

An experimental workflow defines mutations conferring escape from antibody neutralization across the complete CVB3 capsid

The authors begin by studying which mutants escape from monoclonal antibodies with the goal of validating the ability to map escape from antibody neutralization at both the site and mutation level. They selected 3 mutations that are predicted to confer different abilities to escape from neutralization. One mutation has the ability to confer strong escape from neutralization (K227S) and two mutations are predicted to confer low (K650Y) or no (K723E) escape. Why do they use the mutant K227S as the one that confers a strong escape when their study shows that the mutant K227T is the most favored mutation according to the mean mutational differential selection? Moreover, K227T is more favored than K227S in 3 of the 4 replicates.

Mouse polyclonal antibody responses are uniform and discrete.

It is understandable that the authors have not generated a mutant for the positions 654 and 655 (both are top 4 sites that are positively selected) because it is clear with the mutant 650 that there is an antigenic region, but what about mutants in position 800? Is a position that is in the top 4 of positive selected sites, it must be in the analysis of the mutants that escapes from antibodies.

Besides the capacity to escape from K650Y, again the same question, for 2 of the 3 mice the most positive selected mutant is K650C, why do the authors use K650Y?

After seeing figure 3D is clear that all the positive selected positions are in the same region, do have double or triple mutants have the same ability to escape from polyclonal antibodies?

It would be interesting to compare the results of this study to the antigenic structures of other Enteroviruses, such as Poliovirus (DOI: 10.1099/0022-1317-66-5-1159).

Antigenic profiles of human sera are highly variable

For sera h87, how do the authors reconcile the differences between the mutation differential selection and the neutralization of K512L and S513P?

For sera h39, why did the authors choose the mutant P798H? Mutant C320S is more interesting, it is the second most selected mutant at the mutation level.

Escape mutations can be predicted based on structural and evolutionary parameters

Why are mutants in 827 misclassified? How do you define this category?

Finally, it would be interesting to discuss more about the trade-off between fitness and escape capacity. We have recently seen this in real world data with omicron variants of SARS-CoV-2. To this end, I suggest that some of these findings could be tested back in the mouse model instead of just in tissue culture to define the neutralization obtained for the most interesting mutants.

Overall, I found this work significant for the field with remarkable results and rigorously obtained from a methodologically stand point. However, some of those findings should be tested in vivo.

Reviewer #3 (Remarks to the Author):

Advances in genetic and structural techniques have facilitated the study of the interaction of polyclonal responses to several enveloped viruses (e.g. RSV, HIV, Influenza A virus, SARS-CoV-2); however, the

information on non-enveloped viruses is less abundant (e.g. enterovirus A71, norovirus). Alvarez and colleagues used deep mutational scanning to explore the question of how polyclonal sera engaged with the capsid of coxsackievirus B3. The two most significant outcomes of their studies are that despite detecting distinct escape viruses for different individuals, almost all of them targeted an immunodominant epitope on the northern rim of the canyon, and using machine learning (ML), they were able to define key structural and evolutionary parameters that are predictive of escape. The study is of interest for those working in immunity to picornaviruses and other diverse viruses.

Specific comments

1. The title seems to evaluate the complete interaction of polyclonal sera with the viral capsid, but this technology only allows evaluation of neutralizing antibodies. This distinction is quite relevant as only a subset of the antibodies elicited for enteroviruses will have neutralizing activity (Huang et al. Nat Comm 2017), and non-neutralizing antibodies could also confer protection (Yao et al. Hum Vaccin Immunother 2018). Please modify the title and the text to reflect the findings of your study.

2. While the information regarding the polyclonal mapping and immunodominance in non-enveloped viruses is less abundant than for enveloped viruses, authors failed to discuss recent work done for two non-enveloped RNA viruses: enteroviruses A71 (Antanasijevic et al. PNAS Nexus 2022) and noroviruses (Tohma et al. Cell Reports 2022; Lindesmith et al. Viruses 2022).

3. It was hard to follow the differences described for the two groups of 8 initial human sera tested. I suggest providing a graph that summarizes and quantifies all the escape mutations for each of the 8 human sera.

4. Serum samples h4 (aa 512 and 827) and h75 (aa 512 and 829) present common (or close) residues involved in immune escape and similar differential selection values (Figures 3 and 4, Panel A), yet are classified in different antigenic profiles. Based on Panel D from Figures 3 and 4, samples h4 (narrow profile), h39, h19, and h75 (broad profile) present most mutations involved in immune escape at the northern rim of the canyon, so it is not clear why there were classified into “distinct” antigenic profiles. Authors should provide a better explanation on how the different antigenic profiles were defined.

5. Based on the data collected for residues 723 and 798, which are 20A apart, authors argued that antibody responses are focused on multiple epitopes in the northern rim of the canyon. Please elaborate if the possibility that these constitute a single immunodominant epitope in the context of the whole particle was considered. Are residues 723 and 798 (or those in their vicinity) variable in nature? If not, please elaborate on the relevance of this study for enteroviruses vaccine design. In addition, the

numbering the authors use for the different mature proteins does not follow traditional numbering used in the picornavirus field; please consider changing.

6. Despite that the serum samples selected multiple different escape mutants, most of them targeted one single immunodominant region (aa 723 and 798) on the northern rim of the canyon. Please explain why these two mutations were not detected in your initial screenings for samples h87, h46 and h75.

7. Authors discuss that samples with narrow antigenic profiles present, in general (4/5), high titers and are more susceptible to immune evasion, but there are two things to consider in the interpretation of this data: (1) (at least) two of those residues (227 and 473) conferring immune evasion have minimal impact on the virus, and (2) all these high titer sera were susceptible to changes in residues 723 and 798. This suggests that these are co-dominant epitopes and this system just detected those with higher probability of rescue. Authors could perform additional passages with these sera samples to determine if the selection will ultimately lead to viruses with mutations in two distant epitopes.

8. Interestingly, this study shows that the sera samples with high neutralizing activities show mutations at the 3- and 2-fold plateau, but Huang and colleagues (Nat Comm 2017 DOI: 10.1038/s41467-017-00736-9) showed that individuals presented lower titers against recent circulating strains of enterovirus A71. Please consider discussing these differences.

9. Authors argued that differences in the antigenic profile could be due to different history of infection. The existence of cross-reactive antibodies to different enteroviruses in human sera was recently shown (Rosenfeld et al. mBio 2022), so another possibility to consider is that the serum samples with lower neutralizing titer could have neutralization ability against other enteroviruses. This could be easily tested performing neutralization experiments with a few additional enteroviruses B with these samples.

10. The data collected with ML analyses is quite interesting. Why did the authors not try to validate/test their models with data from other enteroviruses with extensive escape mutant information, e.g. EV71 (Huang et al. Nat Comm 2017; Huang et al. Nat Comm 2020)?

11. In their discussion, the authors stated “whether these patterns are conserved in other picornaviruses and/or non-enveloped viruses remains to be elucidated;” however, two recent complementary studies demonstrated the role of antibody immunodominance in the evolution of the non-enveloped norovirus (Tohma et al. Cell Reports 2022; Lindesmith et al. Viruses 2022), and that antigenic differences are mostly due to multiple mutations occurring in different epitopes. This should be considered in their discussions.

12. Please update the reference section with the studies suggested above. Note that reference #15 (#31) and #44 (#48) are duplicated and references 43, 45 and 51 appear incomplete.

A point-by-Point response to the Reviewers' comments

We thank the Reviewers for their helpful comments and suggestions. Below, please find our responses to each point raised in blue font.

Reviewer #1 (Remarks to the Author):

Review of "Comprehensive profiling of polyclonal sera targeting a non-enveloped viral capsid"

This paper uses deep mutational scanning to look at antibody escape mutations from monoclonal antibodies, polyclonal mouse serum, and polyclonal human serum. The experiments work effectively as validated by neutralization assays, and provide new insight into the antigenicity of the CVB3 capsid proteins. The authors also making interesting observations about the variability of specificity among humans (as compared to mice) and the breadth of epitopes targeted relative to other viruses.

Overall, this is an interesting and well done study, and I support its publication with minor comments.

MAJOR COMMENTS:

- It would be helpful to discuss more why human sera have different epitopes? The authors mention in discussion the possibility of variable exposure histories. More elaboration would be useful: for instance, how variable are circulating CVB3-like viruses at these sites?

We thank the Reviewer for this suggestion. The different profiles observed for human sera samples may be explained by variable exposure histories to different enteroviruses since enterovirus infections result in cross-reactive antibodies that can neutralize heterologous enteroviruses (Rosenfeld et al., 2022). Another alternative, as suggested by the Reviewer, is that exposure to different CVB3 variants changes the neutralization profiles since escape sites are highly variable (median Shannon entropy of 1.5 for escape sites within the Enterovirus B family versus 0.65 for surface-exposed sites not harboring escape mutations; $p < 1e-11$) it has been suggested that repeated infections broaden the immune response against related variants in the case of the non-enveloped norovirus (Lindesmith et al., 2022). Finally, genetic differences between humans are also likely to underlie variability in the immune response. We have included these hypotheses in the discussion (lines 521-524 and 528-548) and included the analysis of CVB3 variability (Shannon entropy analysis) in escape vs non-escape surface sites in the supplementary material (see Figure S5D).

- I found the machine learning section less compelling than some of the others. Wildtype amino acid identity helps predict which sites are escape sites, but is this just confounded with surface accessibility: is it surface accessible sites that mediate escape and those tend to be charged / polar? Beyond that, what are we learning from machine learning? If different human sera have different escape mutations then clearly that must be driven by some non-viral factors (that aren't included in the machine learning

model).

Our analysis was done comparing escape sites to other surface-exposed sites in which escape was not observed. Consequently, the properties that help to predict escape are not common to all surface exposed residues but instead are particularly enriched in those surface sites that confer escape versus other surface sites that do not confer escape (see Figure S5). Furthermore, we agree that non-viral factors are likely to have a strong contribution in driving different antigenic profiles. However, the main aim of the machine learning analysis was to determine which combination of features is the most relevant for predicting escape. The fact that we observe relatively high predictability tells us that the identified viral and evolutionary factors are indeed informative for understanding antibody neutralization sites. In regards to whether the nature of the WT and mutant AA is simply reflecting relative surface exposure (or for any of the other individual variables examined), we perform a reshuffling experiment on all variables independently (essentially removing any information from that feature) and reevaluate the ability of the algorithm to predict escape (Figure 5C). The results of this analysis reveal that none of the variables are as informative as WT and the mutation, indicating these capture a combination of attributes that are not provided by other variables, such as RSA.

- Since these experiments create mutants of a human pathogenic virus, some discussion should be given to biosafety implications of making these mutants. Do the experiments combining mutations create mutants with more antigenic escape than likely to arise naturally? How did the authors think about this, and what types of biosafety considerations should apply to this and similar research?

We thank the Reviewer for their comment. Indeed, we have considered these aspects. As RNA viruses rapidly acquire mutations during their replication in nature and can recombine to mix mutations, the introduction of double mutations in a highly prevalent virus such as CVB3 is unlikely to represent a risk beyond that occurring daily in nature in non-controlled settings and in the absence of BSL2 safety measures employed in our lab. As with any of our work with potential for biosafety risks, we operate in a certified BSL2 facility and have obtained permission for all of these experiments from the biosafety committees of both our institute and that of the University of Valencia, as well as the national committee overseeing genetically modified organisms in Spain. We now include a statement about this in the methods (lines 689-692).

MINOR COMMENTS:

- Second paragraph of Results, the text refers to an interactive version of Figure 1F and links to the GitHub. However, when I navigate to the GitHub it is not really clearly documented how to find this particular interactive visualization (I couldn't find it). Maybe a more explicit link could be provided, or more details in the GitHub README? The same point goes for other links to the GitHub throughout the document: it's hard to figure out exactly where in the GitHub the relevant data are found. All of this could probably be fixed with a better README.

We thank the Reviewer for this suggestion and have provided additional details in the readme files of the GitHub repository. Additionally, we have included a specific link in the GitHub folder "dms_view" that

directly leads to the interactive visualization of the escape profiles, as indicated in the manuscript (see lines 138-139 and 776-778).

- Figure 2: the three mice all share the same y-axis scale. Maybe this is reasonable to do, but I would not object if the scales were different for the mice either since often the extent of differential selection in such experiments depends on the serum concentration and IC50 which may differ a bit. Basically, it's fine as is, but each could also be on its own scale. Similar comments apply to some other figures.

We agree with the Reviewer that the scales do not necessarily need to be the same in all samples since the range of differential selection values is determined by the selection strength applied in each particular case. However, we decided to use the same scale for all samples in each figure to help the reader better gauge differences between the individual sera, which may otherwise seem to neutralize to the same degree if we vary the Y-axis. However, we provide the data in the GitHub repository so that interested readers can evaluate the data on their own.

- Is there a biological explanation why we might expect neutralization to be different between male and female humans?

Some studies have shown different adaptive responses between males and females to picornavirus infections (Wang et al., 2006; Wang et al., 2017). Sex-specific hormones (e.g. Klein and Flanagan, 2016) and genetic background (Fish 2008) may influence these responses. Estrogen can enhance B cell activation and antibody production, and testosterone can suppress B cell responses. Some immune-related genes are present in the X chromosome. Aside from these, non-biological reasons for this could also exist (e.g. time spent in close proximity to children). However, we do not address this issue in the current paper and there is no consensus on what could be driving this to the best of our knowledge.

- In Figure S3 and S4, why did replicates per sera differ between individual sera? This is probably fine and I am not asking for more replicates, but just explanation of this fact would be helpful.

We set up the experimental pipeline using the mAb, the mouse sera samples, and 4 of the top neutralizing human samples with additional replicates to assess variability in the experimental approach. Once we analyzed this dataset, we concluded that technical replicates were not needed and therefore did not include these in subsequent analyses (the remaining 4 human sera). The information on the number of replicates for each dataset is included in the methods (lines 757-768).

Reviewer #2 (Remarks to the Author):

Alvarez-Rodriguez et al., obtained the full antigenic profile of multiple human and mouse sera targeting the capsid of a prototypical picornavirus. Their results uncover significant variation in the breadth and strength of neutralization sites targeted by individual human polyclonal responses, which contrasted with homogenous responses observed in experimentally infected mice. Also, they use these comprehensive antigenic profiles to define key structural and evolutionary parameters that are predictive of escape, assess epitope dominance at the population level, and reveal a need for at least

two mutations to achieve significant escape from multiple sera. Overall, data provide the first comprehensive analysis of how polyclonal sera target a non-enveloped viral capsid and help define both immune dominance and escape at the population level in tissue culture.

Minor:

All figures:

Why did the authors use the structure of strain SD (PDB ID:4GB3)?

We only use the structure of strain SD (PDB ID: 4gb3) in Figure 1E to depict the four main antibody binding regions in picornaviruses. The rest of the structures shown in the paper are based on the CVB3 strain SD capsid pentamer mutated to encode the Nancy strain amino acid sequence with FoldX (Schymkowitz et al., 2005) as previously reported ((Mattenberger et al., 2021), see Methods section). We chose this structure since the identity between the capsids of these two strains (SD and Nancy) at the amino acid level is 99% (843 identical residues out of a total of 851), and to facilitate integration with our previous analyses (Mattenberger et al., 2021).

Figure 4c comment:

Please correct the order of mutations for sera h39.

We thank the Reviewer for pointing out this mistake, we have modified Figure 4 accordingly.

Figure 6c comment:

Please include the ND in Figure 6C.

ND is present both in the figure (part of the IC50-fold change) and in the text of Figure 6C.

Page 7 - "Unexpectedly, all three sera showed a similar antigenic profile, suggesting that the response of experimentally immunized mice is relatively invariant..." - Why is this result unexpected given the genetic inbreeding of laboratory strain mice?

Indeed, these mice are genetically identical. However, monoclonal antibodies that target all antigenic regions of the viral capsid have been isolated for picornaviruses, showing mice can mount different responses (Huang, Current Opinion in Virology, 2021). Moreover, cryoEM analysis of polyclonal antibodies from immunized mice bound to the CVA21 capsid showed binding across two epitopes (Antanasijevic et al. 2022). As the model used in the current work involves challenging with a live, replication-competent virus, we anticipated the stochastic events would result in distinct signatures; indeed, we observe different neutralizing titers between the mice, supporting variability in the immune response generated in this mouse model. We have now included a discussion on why we find this unexpected in the discussion (lines 489-493).

REFERENCES

References 15 and 31 are the same one.

We thank the Reviewer for pointing out this error, we have removed the duplicated reference.

Major:

An experimental workflow defines mutations conferring escape from antibody neutralization across the complete CVB3 capsid

The authors begin by studying which mutants escape from monoclonal antibodies with the goal of validating the ability to map escape from antibody neutralization at both the site and mutation level. They selected 3 mutations that are predicted to confer different abilities to escape from neutralization. One mutation has the ability to confer strong escape from neutralization (K227S) and two mutations are predicted to confer low (K650Y) or no (K723E) escape. Why do they use the mutant K227S as the one that confers a strong escape when their study shows that the mutant K227T is the most favored mutation according to the mean mutational differential selection? Moreover, K227T is more favored than K227S in 3 of the 4 replicates.

We proceeded to validate our results once we had the data for some of the human sera. To facilitate our experiments, given the high number of mutants and different sera tested in this study (and taking into account limitations in the quantity of sera available), we selected those mutations in a particular site that conferred strong escape and were present in multiple sera. In the specific case that the Reviewer raises, residue K227, the site is shared between the profiles of the mAb and h8. We chose to use K227S for validation as it represents the shared mutation conferring the strongest escape from both the mAb and h8. This applies to most mutants selected for validation. Additionally, this allowed us to validate mutations having different levels of positive differential selection and not only those with the largest effect.

Mouse polyclonal antibody responses are uniform and discrete.

It is understandable that the authors have not generated a mutant for the positions 654 and 655 (both are top 4 sites that are positively selected) because it is clear with the mutant 650 that there is an antigenic region, but what about mutants in position 800? Is a position that is in the top 4 of positive selected sites, it must be in the analysis of the mutants that escapes from antibodies.

Residue 800 lies structurally in the same antigenic region as the other two validated sites for this serum (650 and 723). We chose to validate the latter mutations rather than residue 800 because site 650 is the main site of escape and site 723 was shared with escape profiles from some human samples (h39, h19).

Besides the capacity to escape from K650Y, again the same question, for 2 of the 3 mice the most positive selected mutant is K650C, why do the authors use K650Y?

In mouse 3 (m3) K650C presents a lower positive differential selection score. For this reason, we picked the mutation that showed a strong positive differential selection score in all 3 mice (K650Y).

After seeing figure 3D is clear that all the positive selected positions are in the same region, do double or triple mutants have the same ability to escape from polyclonal antibodies?

Since the antigenic profiling approach provides information on the effect of single amino acid mutations, we only tested single amino acid changes to validate the profiles. Moreover, in several cases, we obtained strong escape with single mutations and it is not clear we would be able to get better escape in these cases with two mutations. On the other hand, we decided to combine mutations to better understand immune dominance in the larger sample once we identified common epitopes. In this case, we had many sera that did not show strong escape from any of the single mutants assessed. We, therefore, thought it was important to identify in this dataset if double mutants would result in increased escape. Indeed, that is what we observed for residues 723 and 798, which we now believe represent a single epitope (see lines 597-607), supporting the fact that multiple mutations within the same epitope can improve escape.

It would be interesting to compare the results of this study to the antigenic structures of other Enteroviruses, such as Poliovirus (DOI: 10.1099/0022-1317-66-5-1159).

We have now included a supplementary table with the secondary structure elements involved in escape in CVB3 versus other enteroviruses to facilitate such analysis (Table S6) and refer the reader to it in the text (lines 399-401).

Antigenic profiles of human sera are highly variable

For sera h87, how do the authors reconcile the differences between the mutation differential selection and the neutralization of K512L and S513P?

Indeed, the expectation is that escape mediated by L will be larger than that by P (according to the differential selection score). We believe this to be the result of some variability in the data and in this sample in particular, as only a single replicate is being analyzed for this sample (as indicated in the methods). Please note that in the other seven human samples and three mouse samples, consistency between differential selection scores and effect are observed, and that for this sample, both the P and L mutations yield observable escape, validating the sites harboring these mutations as escape sites.

For sera h39, why did the authors choose the mutant P798H? Mutant C320S is more interesting, it is the second most selected mutant at the mutation level.

As indicated above, we chose to only validate some sites in each profile, focusing on mutations shared between sera to reduce the number of assays performed. In the case of h39, we validated a mutation that was common with sera h46.

Escape mutations can be predicted based on structural and evolutionary parameters
Why are mutants in 827 misclassified? How do you define this category?

Misclassified mutations (those we do not experimentally identify as conferring escape but classified as likely to confer escape by the machine learning algorithm) occurred in 12 sites where strong escape was already observed with other mutations in the antigenic profiling experiments, strongly suggesting these are likely to confer escape. The reasons for not observing these mutations could either be due to some

noise in the experiments or that some of these mutations may not be present in our initial populations due to having low fitness, stochastic events, or if they are only 1 nucleotide mutation away from the WT codon, as we exclude single mutations per codon to increase the signal to noise ratio (indicated in the methods).

Finally, it would be interesting to discuss more about the trade-off between fitness and escape capacity. We have recently seen this in real world data with omicron variants of SARS-CoV-2. To this end, I suggest that some of these findings could be tested back in the mouse model instead of just in tissue culture to define the neutralization obtained for the most interesting mutants. Overall, I found this work significant for the field with remarkable results and rigorously obtained from a methodologically stand point. However, some of those findings should be tested in vivo.

This is indeed a very interesting point. We are not sure that fitness in the inbred mouse model can be extrapolated to what is observed in humans, the natural hosts of the virus. However, we are currently planning on performing a study in mice to assess the evolution of immune responses to homologous and heterologous challenges, using the escape mutants defined in this work, where we will also assess the fitness of different escape mutants observed in mice. This is a long-term project that we believe is outside the scope of the current manuscript, which is already quite extensive and with numerous new insights into the interaction of CVB3 with neutralizing antibodies.

Reviewer #3 (Remarks to the Author):

Advances in genetic and structural techniques have facilitated the study of the interaction of polyclonal responses to several enveloped viruses (e.g. RSV, HIV, Influenza A virus, SARS-CoV-2); however, the information on non-enveloped viruses is less abundant (e.g. enterovirus A71, norovirus). Alvarez and colleagues used deep mutational scanning to explore the question of how polyclonal sera engaged with the capsid of coxsackievirus B3. The two most significant outcomes of their studies are that despite detecting distinct escape viruses for different individuals, almost all of them targeted an immunodominant epitope on the northern rim of the canyon, and using machine learning (ML), they were able to define key structural and evolutionary parameters that are predictive of escape. The study is of interest for those working in immunity to picornaviruses and other diverse viruses.

Specific comments

1. The title seems to evaluate the complete interaction of polyclonal sera with the viral capsid, but this technology only allows evaluation of neutralizing antibodies. This distinction is quite relevant as only a subset of the antibodies elicited for enteroviruses will have neutralizing activity (Huang et al. Nat Comm 2017), and non-neutralizing antibodies could also confer protection (Yao et al. Hum Vaccin Immunother 2018). Please modify the title and the text to reflect the findings of your study.

We apologize for not being sufficiently specific. We have now changed the title to “Comprehensive profiling of **neutralizing** polyclonal sera targeting a non-enveloped viral capsid” and corrected this across the text.

2. While the information regarding the polyclonal mapping and immunodominance in non-enveloped viruses is less abundant than for enveloped viruses, authors failed to discuss recent work done for two non-enveloped RNA viruses: enteroviruses A71 (Antanasijevic et al. PNAS Nexus 2022) and noroviruses (Tohma et al. Cell Reports 2022; Lindesmith et al. Viruses 2022).

We thank the Reviewer for bringing this literature to our attention. We have now included a discussion of these results in the discussion section of the manuscript.

3. It was hard to follow the differences described for the two groups of 8 initial human sera tested. I suggest providing a graph that summarizes and quantifies all the escape mutations for each of the 8 human sera.

We apologize for not making this distinction clear enough. It is based on the number of regions targeted in the profile, which is summarized in Figure 6A. We have now included a statement earlier on to help with this distinction (lines 200-201).

4. Serum samples h4 (aa 512 and 827) and h75 (aa 512 and 829) present common (or close) residues involved in immune escape and similar differential selection values (Figures 3 and 4, Panel A), yet are classified in different antigenic profiles. Based on Panel D from Figures 3 and 4, samples h4 (narrow profile), h39, h19, and h75 (broad profile) present most mutations involved in immune escape at the northern rim of the canyon, so it is not clear why there were classified into “distinct” antigenic profiles. Authors should provide a better explanation on how the different antigenic profiles were defined.

We have separated the sera into either a narrow or broad profile based on the number of different sites being targeted (one or two that are strongly targeted in narrow, 3 or more that are weakly targeted in broad; note the scale change between figures 3 and 4). The separation is not based on which epitope is targeted in each serum. In the example provided by the Reviewer, a single region is targeted in h4 whereas multiple regions are targeted in h75. Specifically, all sites in h4 map to sites targeted by mAbs that are escaped by mutations in the canyon (Vogt, M. R. et al, 2020 and He, M. et al, 2021). In the case of h75, on the other hand, we see this region targeted as well as additional residues conferring escape in the outer surface and threefold plateau. Please see Figure 6A for a visual summary of this distinction.

5. Based on the data collected for residues 723 and 798, which are 20A apart, authors argued that antibody responses are focused on multiple epitopes in the northern rim of the canyon. Please elaborate if the possibility that these constitute a single immunodominant epitope in the context of the whole particle was considered.

We appreciate the Reviewer bringing this point to our attention. We have investigated these mutations in more detail in available structures. We found two structures showing monoclonal antibodies that bind the canyon northern rim of CVA6 and Echovirus 30 with homologous residues to 723 and 798 involved in

the Ab footprints (Abs 1D5 and 6C5 in Xu, et al., Nat comm 2017 and Wang, et al., Nat comm 2020, respectively), supporting these two sites belonging to the same epitope. We apologize for not identifying this earlier, and now discuss this finding and its implication in the discussion section.

Are residues 723 and 798 (or those in their vicinity) variable in nature?. If not, please elaborate on the relevance of this study for enteroviruses vaccine design.

Indeed, these residues are variable in nature, forming part of two surface-exposed loops. Some of the flanking residues are also variable. This data is available to the reader in the supplementary table CVB3_nancy_table_escape_muts_sites in GitHub (see entABC column for entropy). Per the suggestion of the Reviewer, we have included a discussion of the implications of this for vaccine design (lines 607-612).

In addition, the numbering the authors use for the different mature proteins does not follow traditional numbering used in the picornavirus field; please consider changing.

In terms of the numbering, we favor the polyprotein nomenclature as we believe it is less confusing for readers outside of the picornavirus field. However, tables S2 and S3 contain both the polyprotein numbering and the chain/residue information for the main sites of escape to facilitate the conversion from one nomenclature to the other. Additionally, we have now included a supplemental table in GitHub that allows for rapid conversion of the polyprotein to chain/residue nomenclature for all residues in the capsid (See CVB3_master_table in the Github folder 10_analysis_sites_muts).

6. Despite that the serum samples selected multiple different escape mutants, most of them targeted one single immunodominant region (aa 723 and 798) on the northern rim of the canyon. Please explain why these two mutations were not detected in your initial screenings for samples h87, h46 and h75.

Our profiling experiments provide information on the effect of single mutations only. We indeed observed site 723 to show escape in all of the mentioned sera [positive site differential selection (Log2) of 3.4, 2.1, and 3.1 for h87, h46, and h75], in agreement with this site being consistently observed in all the sera we tested (see Figure 6C). However, we generally do not observe site 798 having a strong effect on its own (see Figure 6C), but only affecting escape when combined with a mutation in site 723. Indeed, the combined mutations confer stronger escape than the single mutations for all of the mentioned sera (fold-reduction in neutralization of 8.5, 14, and 6.4 for sera h87, h46, and h75). It is for this reason that this site is not observed in the initial screening.

7. Authors discuss that samples with narrow antigenic profiles present, in general (4/5), high titers and are more susceptible to immune evasion, but there are two things to consider in the interpretation of this data: (1) (at least) two of those residues (227 and 473) conferring immune evasion have minimal impact on the virus, and (2) all these high titer sera were susceptible to changes in residues 723 and 798. This suggests that these are co-dominant epitopes and this system just detected those with higher probability of rescue. Authors could perform additional passages with these sera samples to determine

if the selection will ultimately lead to viruses with mutations in two distant epitopes.

We agree with the Reviewer that the sera show co-targeted epitopes, and indeed this is evidenced by the broad sera (Figure 4) and the analysis of immune dominance (Figure 6C). In the validation experiments, we only validated a few sites to provide an experimental context to the results but this does not mean that there are no additional epitopes being targeted (as observed in most profiles). Moreover, we do not believe that we are only observing sites that have low impact on viral fitness (point 1 of the Reviewer above); indeed, several of the mutations show reduced fitness (see Figure 6C). For example, A512L has a strong fitness cost and is observed in 3 sera with escape >7-fold. Moreover, D535T is observed to lead to escape in all sera, despite being deleterious for the virus. Finally, it is important to note that we don't observe a strong peak for some of the narrow sera for the epitope encompassing residues 723/798 in the profile because this site appears to require 2 mutations to show a significant effect. Indeed, we have gone back and tested the effect of single mutations at sites 723 and 798 in the narrow sera h8 (the most neutralizing) to better assess whether these individual mutations show significant contributions to neutralization in this serum. While the dominant site mutation (site 227) results in a nearly 20-fold reduction in neutralization, mutations at 723 or 798 individually do not reach a 2-fold escape, indicating that this epitope plays a non-prominent role in this serum. It is not clear to us that an experimental selection regimen would be able to select mutations at this secondary, non-dominant site with such weak selection pressure while contending with the high likelihood of the emerging virus being low fitness due to having three mutations. Overall, we believe our data shows that the profiling experiments detect both high and low-fitness mutations and can reveal multiple neutralizing epitopes in the polyclonal sera.

8. Interestingly, this study shows that the sera samples with high neutralizing activities show mutations at the 3- and 2-fold plateau, but Huang and colleagues (Nat Comm 2017 DOI: 10.1038/s41467-017-00736-9) showed that individuals presented lower titers against recent circulating strains of enterovirus A71. Please consider discussing these differences.

We thank the Reviewer for bringing this to our attention. Actually, our results are in agreement with this work, where most of the strongly neutralizing sera target the rim and floor of the canyon (e.g. see Figure 6A), and mutations in northern rim residues conferred escape to most of the tested sera; this is in agreement with the mentioned paper (e.g. Huang et al. Nat Comm 2017). We have now added this to the discussion (lines 569-570).

9. Authors argued that differences in the antigenic profile could be due to different history of infection. The existence of cross-reactive antibodies to different enteroviruses in human sera was recently shown (Rosenfeld et al. mBio 2022), so another possibility to consider is that the serum samples with lower neutralizing titer could have neutralization ability against other enteroviruses. This could be easily tested performing neutralization experiments with a few additional enteroviruses B with these samples.

Again, we would like to thank the Reviewer for this suggestion. We have now performed the suggested experiment to see if there could be clear-cut differences in the neutralization of different CVB strains to help better understand the factors contributing to narrow and broad profiles. Unfortunately, we do not see such a difference, suggesting this is not the underlying cause of these differences. We now include this additional data in the manuscript (Figure S7) and in the discussion (lines 548-551).

10. The data collected with ML analyses is quite interesting. Why did the authors not try to validate/test their models with data from other enteroviruses with extensive escape mutant information, e.g. EV71 (Huang et al. Nat Comm 2017; Huang et al. Nat Comm 2020)?

We implemented the ML analysis to help understand better the factors that define escape mutations and how they interact. To this end, we successfully define these factors and show that they are informative for predicting escape. In terms of applying this approach to additional viruses as suggested by the Reviewer, this is an ongoing project in the lab (e.g. for PV, HRV, EVA71). However, this requires a large amount of additional work, obtaining the parameters for each of the viruses, optimizing the standardization methodology between the different viruses so as to be able to extrapolate the results (e.g. differences in particle size, amino acid composition, or B-factor can vary between the viruses), validating them experimentally, and developing a tool so the same can be applied to additional viruses. This is an ongoing project in the lab and we believe this is outside the scope of the current paper, as it implies a large amount of additional data and analyses in an already quite extensive and detailed manuscript.

11. In their discussion, the authors stated “whether these patterns are conserved in other picornaviruses and/or non-enveloped viruses remains to be elucidated;” however, two recent complementary studies demonstrated the role of antibody immunodominance in the evolution of the non-enveloped norovirus (Tohma et al. Cell Reports 2022; Lindesmith et al. Viruses 2022), and that antigenic differences are mostly due to multiple mutations occurring in different epitopes. This should be considered in their discussions.

We thank the Reviewer for pointing out these studies. We have now included a discussion of these results in the discussion section (see lines 500-502, 521-524).

12. Please update the reference section with the studies suggested above. Note that reference #15 (#31) and #44 (#48) are duplicated and references 43, 45 and 51 appear incomplete.

Thank you for identifying these errors. We have corrected this.

REVIEWER COMMENTS

Reviewer #1 (Remarks to the Author):

The revised version has satisfactorily addressed the critiques.

Reviewer #2 (Remarks to the Author):

I now think that the manuscript has improved considerably, and I would like to see it published. However, I believe that some experiments, which I do not think are difficult to perform, would be necessary to address the following point:

You choose a mAb to validate your approach. It is understandable that it is not possible to test the sera against all possible escape mutants, but in this case it is a commercial mAb. You are validating your data with the second best option. Why not the first; why not generate this escape mutant? By generating a mutant that you already know is viable, this is not that hard to verify and you would save the reader a lot of questions. Based on this observation, how well does this mAb neutralize the viral infection? If so, how potent is it? Since this is a validation, I would consider a broader approach where you try at least 3 different positions and 3 different mutants to validate your results? For example, positions 227, 242, and 396. In all cases you could have mutants that escape, low escape, and no escape.

Reviewer #3 (Remarks to the Author):

Thank you for addressing most of my comments. This reviewer still have doubts about the metrics that the authors used to separate the serum samples into two distinct antigenic profiles. Authors distinguished these profiles based on the impact that single mutations have on neutralization (IC50) titers. While there are two serum samples (h8 and h112, probably exceptions) that show major IC50 changes (>16 fold IC50 change), the two serum samples (h4 and h87) included in the "narrow profile" group presented several mutations with <16 fold change in IC50 titer (Figures 3C and 6). Thus, the serum h4 and h87 seems to present a "broad profile" as most of the sera tested (Figure 6).

Since (i) the majority of the sera tested presented minor changes (<10 fold) in IC50 titer when only one mutation tested (Figures 3, 4, and 6), (ii) most mutations have a major effect in viral fitness, and (iii) two mutations (e.g. 723 and 798) seems to be required to have a major effect in most samples, the overall conclusion seems to be that most sera target multiple co-dominant epitopes that comes with a fitness "price" and precludes antigenic evolution of CVB3. This reviewer believes that a better emphasis should be made on this regard in the discussion section.

Specific comments:

1. In your abstract you stated "the first comprehensive analyses of how polyclonal sera target a non-enveloped viral capsid" but there is previous data on EV71 and norovirus that addressed this topic. Please modify.
2. It is unfortunate that the authors did not test the higher neutralizing serum samples with all the mutations to determine their overall role in IC50 titer change (Figure 6). A 5-20 fold change in the IC50 means that there is still neutralization activity in the sera and the overall role of the different epitopes could provide insights on their immunodominance.
3. Lines 483-485: Please consider citing the work done with polyclonal sera in EV71 and norovirus

here.

4. Lines 490-493: This reviewer is uncertain why this is unexpected for the authors, as (i) mAb development disentangle polyclonal responses, and (ii) you captured a reduction of ~20 fold increase during a single round of replication, thus selecting the most immunodominant epitope. Further passages under high sera concentration might select for other less immunodominant epitopes that are detected at monoclonal level. This should be discussed as limitation rather than show "surprise" that their system did not detect these epitopes.

5. Line 572: Please change K277S to K227S.

Response to referees:

Reviewer #1 (Remarks to the Author):

The revised version has satisfactorily addressed the critiques.

Reviewer #2 (Remarks to the Author):

I now think that the manuscript has improved considerably, and I would like to see it published. However, I believe that some experiments, which I do not think are difficult to perform, would be necessary to address the following point:

You choose a mAb to validate your approach. It is understandable that it is not possible to test the sera against all possible escape mutants, but in this case it is a commercial mAb. You are validating your data with the second best option. Why not the first; why not generate this escape mutant?

By generating a mutant that you already know is viable, this is not that hard to verify and you would save the reader a lot of questions. Based on this observation, how well does this mAb neutralize the viral infection? If so, how potent is it?

Since this is a validation, I would consider a broader approach where you try at least 3 different positions and 3 different mutants to validate your results? For example, positions 227, 242, and 396. In all cases you could have mutants that escape, low escape, and no escape.

As per the Reviewer's suggestion, we have now generated an additional 8 mutants from the main sites of escape. Accordingly, we now provide data for 9 mutants conferring escape (3 each from sites 227, 242, and 396) that include the top mutants in each site, and 2 mutants predicted to not confer escape across 2 different sites (Figure 1C,D). All mutants tested from the main escape sites 227, 242, and 396 (all having positive differential selection scores), conferred significant escape from this mAb, validating the relevance of these sites to escape. In contrast, the mutants from control sites did not significantly alter neutralization. This data is now included in the manuscript (Figure 1C,D) and the supplementary table (https://github.com/RGellerLab/CVB3-Antigenic-Profiling/blob/main/8_neutralization_results/ic50_results_mice.csv).

In addition, we now include the neutralization titer of the mAb in the manuscript text, as requested. This mAb has an IC50 of 1:5,525, in agreement with the manufacturer's specifications (<https://www.sigmaaldrich.com/ES/en/product/mm/mab948>).

Reviewer #3 (Remarks to the Author):

Thank you for addressing most of my comments. This reviewer still have doubts about the metrics that the authors used to separate the serum samples into two distinct antigenic profiles. Authors distinguished these profiles based on the impact that single mutations

have on neutralization (IC50) titers. While there are two serum samples (h8 and h112, probably exceptions) that show major IC50 changes (>16 fold IC50 change), the two serum samples (h4 and h87) included in the "narrow profile" group presented several mutations with <16 fold change in IC50 titer (Figures 3C and 6). Thus, the serum h4 and h87 seems to present a "broad profile" as most of the sera tested (Figure 6).

Since (i) the majority of the sera tested presented minor changes (<10 fold) in IC50 titer when only one mutation tested (Figures 3, 4, and 6), (ii) most mutations have a major effect in viral fitness, and (iii) two mutations (e.g. 723 and 798) seems to be required to have a major effect in most samples, the overall conclusion seems to be that most sera target multiple co-dominant epitopes that comes with a fitness "price" and precludes antigenic evolution of CVB3. This reviewer believes that a better emphasis should be made on this regard in the discussion section.

We thank the reviewer for their suggestions. We have taken several steps to address the concerns raised.

1. We have now included a quantitative description of why we have separated the sera into broad and narrow profiles. This is explained in the main text (lines 198-203) and supplementary figure S2C. It shows the distinction that a single site dominates escape in narrow profile sera, accounting for >50% of the total positive differential selection, while the maximum escape conferred by any given site in the broad sera is <10% of the total.
2. We have made it clearer that escape from neutralization by single mutations is observed largely in 2 of the narrow profiles (h8 and h112, lines 500-502).
3. We have added a discussion of the limitation of our technique to identify only single mutations, the value of assessing double mutations, and the potential bias towards identifying more dominant or co-dominant epitopes of the technique. In addition, we present future research lines that can address this topic based on our findings (lines 588-601).
4. As we observe that the double mutant (723/728) does not show very low fitness compared to the WT virus, and we have not extensively assayed combinations of two mutations in terms of escape and fitness, we are not comfortable making a speculation that this phenomenon prevents antigenic evolution of CVB3.

Specific comments:

1. In your abstract you stated "the first comprehensive analyses of how polyclonal sera target a non-enveloped viral capsid" but there is previous data on EV71 and norovirus that addressed this topic. Please modify.

While we agree, and repeatedly cite in the paper, that there is existing data that provides insights into how picornaviruses and other non-enveloped viruses escape neutralizing antibodies, we believe that our study is much more exhaustive and of broader scope. Largely, prior studies focus on the characterization of escape for a few mutations and sites. In contrast, our analysis is much more comprehensive, providing a detailed picture of how neutralizing antibodies target the full capsid region (rather than selecting the dominant escape mutation). Nevertheless, we have toned down this statement in the abstract to address the Reviewer's concern.

2. It is unfortunate that the authors did not test the higher neutralizing serum samples with all the mutations to determine their overall role in IC50 titer change (Figure 6). A 5-20 fold change in the IC50 means that there is still neutralization activity in the sera and the overall role of the different epitopes could provide insights on their immunodominance.

We did not test these sera as we obtained a more detailed picture of how these sera are escaped using the antigenic profiling experiments. We tested a few mutants for each of the remaining 18 sera because we did not have the profiling information for these. The highly neutralizing sera are included in the table for comparison.

3. Lines 483-485: Please consider citing the work done with polyclonal sera in EV71 and norovirus here.

We have altered this sentence to include citations and to indicate that there is prior knowledge but a lack of global resolution into these responses in polyclonal sera.

4. Lines 490-493: This reviewer is uncertain why this is unexpected for the authors, as (i) mAb development disentangle polyclonal responses, and (ii) you captured a reduction of ~20 fold increase during a single round of replication, thus selecting the most immunodominant epitope. Further passages under high sera concentration might select for other less immunodominant epitopes that are detected at monoclonal level. This should be discussed as limitation rather than show "surprise" that their system did not detect these epitopes.

We have removed the sentence about the results in mice being unexpected and have now included a few sentences about the limitations of our study to detect subdominant and double mutations, as well as alternative approaches to overcome these limitations in future work (see the last paragraph of the Discussion).

5. Line 572: Please change K277S to K227S.

Thank you for noticing this typo. We have corrected the text accordingly.

REVIEWERS' COMMENTS

Reviewer #2 (Remarks to the Author):

This new version has now addressed all my concerns. Thanks

Reviewer #3 (Remarks to the Author):

The authors have now addressed all my suggestions. Great study!